# ADG: Ambient Diffusion-Guided Dataset Recovery for Corruption-Robust Offline Reinforcement Learning

**Zeyuan Liu[1]\*, Zhihe Yang[2]\*, Jiawei Xu[3]\*, Rui Yang[4], Jiafei Lyu[1], Baoxiang Wang[3],**
**Yunjian Xu[2]†, Xiu Li[1]†**
[1]Tsinghua Shenzhen International Graduate School, Tsinghua University,
[2]The Chinese University of Hong Kong,
[3]The Chinese University of Hong Kong, Shenzhen,
[4]University of Illinois Urbana-Champaign

## Abstract

Real-world datasets collected from sensors or human inputs are prone to noise and errors, posing significant challenges for applying offline reinforcement learning (RL). While existing methods have made progress in addressing corrupted actions and rewards, they remain insufficient for handling corruption in high-dimensional state spaces and for cases where multiple elements in the dataset are corrupted simultaneously. Diffusion models, known for their strong denoising capabilities, offer a promising direction for this problem—but their tendency to overfit noisy samples limits their direct applicability. To overcome this, we propose **A**mbient **D**iffusion-**G**uided Dataset Recovery (**ADG**), a novel approach that pioneers the use of diffusion models to tackle data corruption in offline RL. First, we introduce Ambient Denoising Diffusion Probabilistic Models (DDPM) from approximated distributions, which enable learning on partially corrupted datasets with theoretical guarantees. Second, we use the noise-prediction property of Ambient DDPM to distinguish between clean and corrupted data, and then use the clean subset to train a standard DDPM. Third, we employ the trained standard DDPM to refine the previously identified corrupted data, enhancing data quality for subsequent offline RL training. A notable strength of ADG is its versatility—it can be seamlessly integrated with any offline RL algorithm. Experiments on a range of benchmarks, including MuJoCo, Kitchen, and Adroit, demonstrate that ADG effectively mitigates the impact of corrupted data and improves the robustness of offline RL under various noise settings, achieving state-of-the-art results. Our code is available at `https://github.com/sand-nine/ADG`.

## 1 Introduction

Offline reinforcement learning (RL) has emerged as a prominent paradigm for learning decision-making policies from offline datasets [20, 11]. Existing approaches can be broadly categorized into MDP-based methods [11, 10, 18, 19, 4, 5, 12] and non-MDP methods [6, 16, 29]. However, due to the data-dependent nature of offline RL, it encounters significant challenges when dealing with offline data subjected to random noise or adversarial corruption [42, 41, 22, 36, 37]. Such disturbances can cause substantial performance degradation or result in a pronounced deviation from

---

\*Equal contribution
†Corresponding author

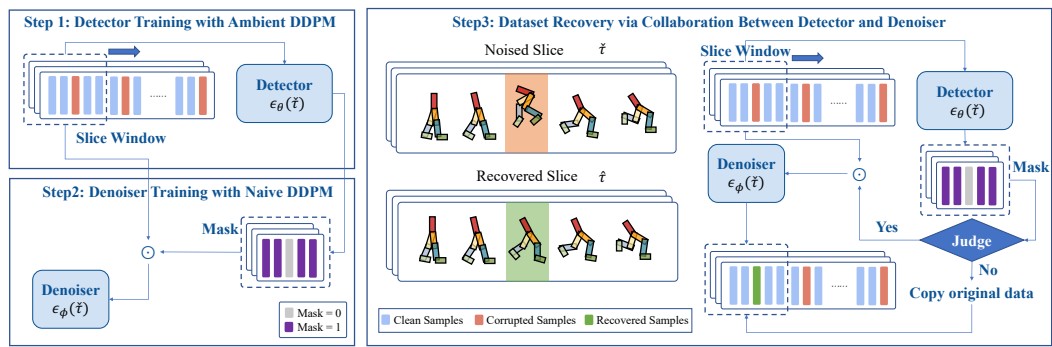

Figure 1: Overview of the training processes for the detector and denoiser (left) and the dataset recovery process (right) in the proposed ADG method.

the intended policy objectives. Therefore, ensuring robust policy learning is crucial for offline RL to function effectively in real-world scenarios.

Several previous studies have focused on the theoretical properties and certification of offline RL under corrupted data [43, 40, 33, 7, 39]. Empirical efforts have resulted in the development of uncertainty-weighted algorithms utilizing Q-ensembles [40] and robust value learning through Huber loss and quantile Q estimators [37]. Additionally, sequence modeling techniques have been applied to mitigate the effects of data corruption while iteratively refining noisy actions and rewards in the dataset [34]. However, as noted in [34], recovering observations remains challenging due to the high dimensionality.

In recent years, diffusion-based generative models [30, 14, 31] have gained considerable attention for their ability to effectively model complex data distributions, making them increasingly important in offline RL [15, 3, 23]. One promising application of these models is their usage in reducing or mitigating noise in data, thanks to their inherent denoising capabilities. For instance, DMBP [38] proposed a diffusion-based framework aimed at minimizing noise in observations during the testing phase. Despite its success, this approach is specifically designed to work with clean training data and only addresses data perturbations during testing. It cannot manage perturbations in the training dataset, as current diffusion models typically assume the dataset is entirely clean or has a consistent noise distribution across all data points [1, 8]. Therefore, naive diffusion methods can encounter challenges, such as overfitting to noise data, when applied to partially corrupted data during training.

To gain deeper insight into which aspects of corrupted datasets degrade the performance of offline RL algorithms, we evaluate several offline RL methods on three types of datasets: a clean dataset, a partially corrupted dataset, and a filtered dataset, which is created by removing the noisy portions from the corrupted data, as illustrated in Figure 2 (a–c). Surprisingly, We find that current offline RL algorithms fail to fully restore their performance on the filtered dataset, particularly when the dataset size is limited. These results imply that the loss of critical sequential information plays a key role in performance degrading of decision-making. Additionally, while sequence modeling methods [6, 34] exhibit robustness against data corruption, they fail to function when faced with incomplete trajectories. **This finding indicates that simply filtering out noisy samples is insufficient for handling corrupted datasets, underscoring the importance of recovering corrupted data.**

To address this issue, *we introduce the first diffusion-based denoising framework for handling corruption robust offline RL.* Our approach recovers clean data purely from the corrupted dataset, without requiring any external information or supervision. We name this novel three-stage diffusion-based method **A**mbient **D**iffusion-**G**uided Dataset Recovery (**ADG**). The detailed diagram is presented in Figure 1. In the first stage, we introduce Ambient Denoising Diffusion Probabilistic Models (DDPM) from *approximated distributions*, which enable diffusion training on partially corrupted datasets with theoretical guarantees. In the second stage, leveraging the noise-prediction property of the well-trained ambient DDPM, we identify the corrupted data within the dataset. The remaining clean data is then used to train a denoiser within the framework of standard DDPM. Finally, in the third stage, we apply the standard DDPM to refine the previously identified corrupted data. The corrected data is combined with the clean data to form a high-quality dataset, which is subsequently used for offline RL training.

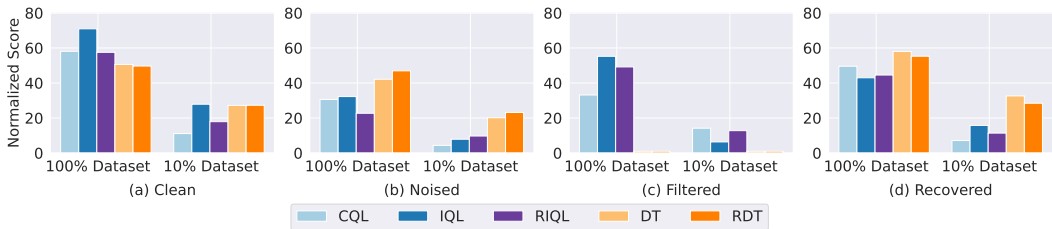

Figure 2: The performance of baseline algorithms (CQL, IQL, RIQL, DT, and RDT) is evaluated under four dataset conditions: **Clean** (original dataset without corruption), **Noised** (dataset corrupted using Random State Attack as described in Appendix C.1), **Filtered** (Noised dataset with corrupted samples removed), and **Recovered** (dataset recovered using ADG). The comparison results show the average normalized scores across three MuJoCo tasks (*halfcheetah*, *hopper*, and *walker2d*) using the "medium-replay-v2" datasets. We include the results for both the 100% dataset and 10% dataset settings. ADG effectively restores performance close to the results on clean datasets.

We find that ADG is overall competitive with the ideal case of perfectly filtering out all noisy samples for MDP-based algorithms and significantly outperforms the filtering method for sequence modeling methods, as shown in Figure 2(d). Additionally, we provide a comprehensive analysis of ADG's performance under both random and adversarial data corruption scenarios, examining various levels of data availability, whether full or limited dataset size, as described in Section 5. Our empirical studies demonstrate that ADG consistently enhances the performance of all baseline algorithms, achieving a remarkable overall improvement. These findings indicate that ADG is versatile and compatible with any offline RL approach, including their robust variants. Notably, ADG exhibits consistent and robust performance across a variety of dataset qualities, corrupted scales, and corrupted ratios.

## 2   Related Works

**Robust Offline RL.**   Several works have focused on testing-time robustness against environment shifts [28, 35, 26, 38, 34]. For training-time robustness, Li et al. [21] investigate various reward attack strategies in offline RL and reveal that certain biases can unintentionally enhance robustness to reward corruption. Wu et al. [33] introduce a certification framework to determine the tolerable number of poisoning trajectories based on different certification criteria. From a theoretical perspective, Zhang et al. [43] propose a robust offline RL algorithm utilizing robust supervised learning oracles. Ye et al. [40] introduce uncertainty weighting to address reward and dynamics corruption, offering theoretical guarantees. Ackermann et al. [2] develop a contrastive predictive coding-based approach to tackle non-stationarity in offline RL datasets. Yang et al. [37] utilize the Huber loss to manage heavy-tailedness and adopt quantile estimators to balance penalization for corrupted data. Additionally, Xu et al. [34] introduce a sequential modeling method to iteratively correct corrupted data for offline RL.

**Diffusion Models in Offline RL.**   Diffusion-based generative models [30, 14, 31] have been extensively utilized for synthesizing high-quality images from text descriptions [27]. More recently, they have gained significant attention in the RL community, serving as behavior policy replicators [32, 13], trajectory generators [15, 3, 23], and state denoisers [38].

## 3   Preliminaries

**RL and Offline RL.**   Reinforcement Learning (RL) is typically formulated as a Markov Decision Process (MDP) defined by the tuple $(S, A, P, r, \gamma)$, where $S$ and $A$ represent the state and action spaces, $P$ denotes the transition function, $r$ is the reward function, and $\gamma \in [0, 1]$ is the discount factor. The objective of RL is to learn a policy $\pi(a|s)$ that maximizes the expected cumulative return. In offline RL, access to the online environment is restricted. Instead, the objective is to optimize the RL objective using a previously collected dataset, $\mathcal{D} = \left\{ \left( s_t^i, a_t^i, r_t^i, s_{t+1}^i \right) \right\}_{i=0}^{N-1}$ which consists of $N$ transitions in total.

**Corruption-robust Offline RL.** We adopt a unified trajectory-based storage approach, as proposed in prior works [34]. An original trajectory is represented as $\tau = (s_0, a_0, r_0, \ldots, s_{T-1}, a_{T-1}, r_{T-1})$, where each trajectory consists of three components: states, actions, and rewards. This trajectory can be reorganized into sequence data for DT [6] and RDT [34], or split into transitions $(s_t, a_t, r_t, s_{t+1})_{t=0}^{T-2}$ for Markov Decision Process (MDP)-based methods such as CQL [19] and IQL [18].

We investigate the impact of injecting random or adversarial noise into the dataset under two corruption scenarios. First, we examine *state* corruption, where only the states in the trajectories are affected. Second, we introduce noise into state-action-reward triplets, which we refer to as *"full-element"* in the following context.

Random corruption refers to the addition of noise drawn from a uniform distribution. For example, corrupting the state with uniform noise of corruption scale $\alpha$ can be written as $\hat{s}_0 = s_0 + \lambda \cdot \text{std}(s), \quad \lambda \sim \text{Uniform}[-\alpha, \alpha]^{d_s}$, where $d_s$ is the dimensionality of the state, and $\text{std}(s)$ represents the $d_s$-dimensional standard deviation of all states in the offline dataset. On the other hand, adversarial corruption employs a Projected Gradient Descent (PGD) attack [25] with pre-trained value functions. We build upon prior work [34] by introducing learnable noise to the target elements and optimizing it through gradient descent to minimize the pretrained value functions. Further details refer to Appendix C.1.

**Diffusion Models.** Given any clean sample $\boldsymbol{x}$, the *forward process* of diffusion models is a Markov chain that gradually adds Gaussian noise to data according to a variance schedule $\alpha_1, \ldots, \alpha_K$:

$$q(\boldsymbol{x}^{1:K} \mid \boldsymbol{x}^0) := \prod_{k=1}^{K} q(\boldsymbol{x}^k \mid \boldsymbol{x}^{k-1}), \quad q(\boldsymbol{x}^k \mid \boldsymbol{x}^{k-1}) := \mathcal{N}(\boldsymbol{x}^k; \sqrt{\alpha_k}\boldsymbol{x}^{k-1}, (1-\alpha_k)\boldsymbol{I}). \quad (1)$$

The *reverse process* is likewise a Markov chain characterized by learned Gaussian transitions (parameterized by $\theta$), typically initiated at $p(x^K) = \mathcal{N}(x^K; 0, \boldsymbol{I})$:

$$p_\theta(\boldsymbol{x}^{0:K}) := p(\boldsymbol{x}^K) \prod_{k=1}^{K} p_\theta(\boldsymbol{x}^{k-1} \mid \boldsymbol{x}^k), \quad p_\theta(\boldsymbol{x}^{k-1} \mid \boldsymbol{x}^k) := \mathcal{N}(\boldsymbol{x}^{k-1}; \boldsymbol{\mu}_\theta(\boldsymbol{x}^k, k), \boldsymbol{\Sigma}_\theta(\boldsymbol{x}^k, k)).$$
$$(2)$$

To ensure high-quality generation, the diffusion model learning process typically demands clean original data, i.e., we have direct access to $\boldsymbol{x}$. Building on the foundation of initial Denoising Diffusion Probabilistic Models (DDPM) introduced by Ho et al. [14], the most widely adopted training loss for diffusion models is formulated as:

$$\mathcal{L}_{\text{DDPM}}(\theta) := \mathbb{E}_{k \sim [1,K], \boldsymbol{\epsilon} \sim \mathcal{N}(\boldsymbol{0}, \boldsymbol{I})}[\|\boldsymbol{\epsilon}_\theta(\boldsymbol{x}^k, k) - \boldsymbol{\epsilon}\|^2], \quad (3)$$

where $\boldsymbol{x}^k = \sqrt{\bar{\alpha}_i}\boldsymbol{x} + \sqrt{1 - \bar{\alpha}_i}\boldsymbol{\epsilon}$ with $\bar{\alpha}_k := \prod_i^k \alpha_i$, and $\boldsymbol{\epsilon}$ is a randomly sampled Gaussian noise. Nevertheless, obtaining completely clean data poses a significant challenge in certain circumstances, making the loss in Eq. (3) not directly applicable. Recently, several studies have proposed methods for handling corrupted datasets, while they assume a uniform noise scale across the dataset [1, 8], leaving the problem of training diffusion models on partially corrupted datasets unresolved.

## 4 Methodologies

### 4.1 Motivation

A natural approach to addressing the partially corrupted dataset is to identify the corrupted samples within the dataset and leverage the uncorrupted samples to train a denoiser, which can subsequently be used to correct the corrupted samples. However, naive noise detectors trained with supervised methods relying solely on manually labeled clean and noisy samples may easily fail due to the inherent ambiguity of noise.

As an alternative, diffusion models have garnered significant attention due to their demonstrated effectiveness in training on fully corrupted datasets and leveraging the trained models to extract uncorrupted samples [1, 8]. Moreover, diffusion models have also been shown to be effective as state denoisers in the realm of offline RL [38].

Nevertheless, a key challenge is that existing diffusion models for corrupted samples typically assume a consistent noise distribution across the dataset, which does not hold for corruption-robust offline RL. On the other hand, if the naive diffusion process is applied to corrupted samples during training, even small amounts of noise in the training set can cause the diffusion model to overfit to these perturbed points, leading to poor performance during subsequent sampling. *To address the discrepancy between clean and corrupted data within the same dataset, the development of a new diffusion training method is urgently needed.*

## 4.2 Diffusion for Partially Corrupt data

**Extending Ambient Diffusion Models to DDPM.** Daras et al. [8] demonstrate that diffusion models can be trained on datasets corrupted by a consistent scale of noise in the context of score-based continuous diffusion models. We observe that their conclusions can be seamlessly extended to the discrete DDPM framework. Building upon Theorem A.5 in [8], which addresses score-based variance preserving diffusion, we derive the following corollary that extends the result to the discrete DDPM.

**Corollary 4.1.** *(Ambient DDPM) Let $k_a$ be a manually defined constant representing a noise-added diffusion timestep. Suppose we are given samples $x^{k_a} = \sqrt{\bar{\alpha}_{k_a}}x^0 + \sqrt{1 - \bar{\alpha}_{k_a}}\epsilon$. Let $x^k = \sqrt{\bar{\alpha}_k}x^0 + \sqrt{1 - \bar{\alpha}_k}\epsilon$ be further en-noised samples with $1 \leq k_a < k$. Then, the unique minimizer of the objective*

$$\mathbb{E}_{x^{k_a}}\mathbb{E}_{k \sim \mathcal{U}(k_a, K)}\mathbb{E}_{x^k | x^{k_a}}\left[\|\frac{\bar{\alpha}_{k_a}}{\sqrt{\bar{\alpha}_k \cdot \bar{\alpha}_{k_a}}}x^k - \frac{(\bar{\alpha}_{k_a} - \bar{\alpha}_k)\epsilon_\theta(x^k, k)}{\sqrt{\bar{\alpha}_{k_a} \cdot \bar{\alpha}_k \cdot (1 - \bar{\alpha}_k)}} - x^{k_a}\|^2\right] \tag{4}$$

*have $\epsilon_{\theta^*}(x^k, k) = \mathbb{E}[\epsilon | x^k], \forall k \geq k_a$ (cf. Appendix A.1 for detailed proof).*

Notably, Eq. (4) cannot be directly applied in our settings because the noise-consistent data, $x^{k_a}$, are not directly accessible. As discussed in Section 5.1, some portions of our dataset are corrupted while others remain clean.

**Ambient DDPM from Approximated Distribution.** It is worth noting that all samples involved in the Ambient DDPM training process follow Gaussian distributions with parameters that are functions of $k$. While direct access to noise-consistent data is not available, an ideal alternative is to approximate these distributions and train the diffusion model using the approximations. Let $q(x^k | x^0)$ denote the ground-truth distribution of samples generated by the DDPM forward process at diffusion timestep $k$, as described in Eq. (1), and let $\varrho(x^k | x^0)$ represent an approximation of this distribution. To ensure effective learning under this approximation, we make the following.

**Assumption 4.2.** *There exists a positive constant $c$ such that, for any $k \geq k_a$, if the Kullback-Leibler (KL) divergence satisfies $\mathbb{D}_{KL}[q(x^k | x^0) \| \varrho(x^k | x^0)] < c$, then the ambient DDPM with $k \geq k_a$, as introduced in Corollary 4.1, can be effectively learned from samples drawn from the approximated distribution $\varrho(x^k | x^0)$.*

Following the standard setup of corruption-robust offline RL, we are provided with samples $\check{x}$ that may or may not contain scaled Gaussian noise $\iota \cdot \epsilon$. The distribution of such samples at diffusion timestep $k$ in the DDPM forward process is denoted as $q(\check{x}^k | \check{x}^0)$. We have the following.

**Theorem 4.3.** *Let Assumption 4.2 hold. For any bounded noise scale $\iota$, one can always find a diffusion timestep $k_a$ such that ambient DDPM with $k \geq k_a$, which should have been learned from samples drawn from $q(x^k | x^0)$, can instead be effectively learned from samples drawn from $q(\check{x}^k | \check{x}^0)$. That is, for any $c > 0$, one can always find $k_a$ such that for any $k \geq k_a$, the following inequality holds: $\mathbb{D}_{KL}[q(x^k | x^0) \| q(\check{x}^k | \check{x}^0)] < c$.*

We provide the detailed proof in Appendix A.2. It is evident that a smaller $k_a$ retains more of the original information, thereby improving the accuracy of the noise predictor. However, a smaller $k_a$ also introduces a more relaxed threshold $c$, which may result in a larger discrepancy between the original distribution and the approximated distribution. The choice of $k_a$ necessitates a trade-off between these two factors. See Section 5.3 for detailed ablation studies.

## 4.3 Corrupted Samples Detection

Given a sample $\check{x}$ that may or may not contain scaled Gaussian noise $\iota \cdot \epsilon$, we propose using the squared Frobenius norm of the noise predictor, $\|\epsilon_\theta(\check{x}, k)\|_F^2$, to determine whether noise is present.

**Proposition 4.4.** *Assume the noise prediction error for $\epsilon_\theta(\cdot, k)$ follows $\delta_\theta^k \sim \mathcal{N}(\mathbf{0}, \sigma_k^2 \mathbf{I})$. Define the difference between the noisy and noise-free cases as: $\Delta = \mathbb{E}[\|\epsilon_\theta(\check{x}_{ns}^k, k)\|_F^2] - \mathbb{E}[\|\epsilon_\theta(\check{x}_{nf}^k, k)\|_F^2]$, where $\check{x}_{ns}^k$ and $\check{x}_{nf}^k$ denote noisy and noise-free samples with original information consistency operation, respectively. The Signal-to-Noise Ratio (SNR) of the prediction is then expressed as:*

$$\mathrm{SNR}(k) := \frac{\Delta}{\mathbb{E}[\|\epsilon_\theta(\check{x}_{nf}, k)\|_F^2]} = \frac{\iota^2 \cdot \bar{\alpha}_k}{(1 - \bar{\alpha}_k) \cdot \sigma_k^2}. \tag{5}$$

See Appendixa A.3 for detailed proof. Assume that the noise prediction error are the same across all diffusion timesteps, i.e., $\sigma_k = \sigma$ for any $k$, then $\mathrm{SNR}(k)$ achieves maximum value at $k = k_a$, as $\bar{\alpha}_k$ is a strictly monotonically decreasing function of $k$.

## 4.4 Ambient Diffusion-Guided Dataset Recovery (ADG)

Having established the theoretical foundation, we now proceed to introduce our proposed ADG method for corruption-robust offline RL. As there are two timesteps involved, we use superscripts $k$ to denote the diffusion timesteps and subscripts $t$ to denote the RL timesteps for clarity.

The ground truth trajectory matrix is defined as $\boldsymbol{\tau}_t := [\boldsymbol{z}_{t-H}, \ldots, \boldsymbol{z}_{t+H}] \in \mathbb{R}^{M \times (2H+1)}$, where $\boldsymbol{z} \in \mathbb{R}^M$ represents the RL component (which can correspond to an observation $\boldsymbol{s}$ or a state-action-reward triplet $(\boldsymbol{s}, \boldsymbol{a}, r)$), $M$ denotes the dimensionality of $\boldsymbol{z}$, and $H$ specifies the temporal slice size. Let $\check{\boldsymbol{z}}_t$ denote the RL component that may or may not contain scaled noise $\iota \cdot \epsilon$. We only have access to the observed (partially corrupted) trajectory $\check{\boldsymbol{\tau}}_t = [\check{\boldsymbol{z}}_{t-H}, \ldots, \check{\boldsymbol{z}}_{t+H}] \in \mathbb{R}^{M \times (2H+1)}$.

**Ambient DDPM for Corrupted Samples Detection.** Following Theorem 4.3, given a partially corrputed offline RL datastet, we pre-define $k_a$ and train the ambient DDPM through

$$\mathbb{E}_{\check{\boldsymbol{\tau}}_t^{k_a} \sim \mathcal{D}, k \sim \mathcal{U}(k_a, K), \check{\boldsymbol{\tau}}_t^k | \check{\boldsymbol{\tau}}_t^{k_a}} \left[ \| \frac{\bar{\alpha}_{k_a}}{\sqrt{\bar{\alpha}_k \cdot \bar{\alpha}_{k_a}}} \check{\boldsymbol{\tau}}_t^k - \frac{(\bar{\alpha}_{k_a} - \bar{\alpha}_k)\epsilon_\theta(\check{\boldsymbol{\tau}}_t^k, k)}{\sqrt{\bar{\alpha}_{k_a} \cdot \bar{\alpha}_k \cdot (1 - \bar{\alpha}_k)}} - \check{\boldsymbol{\tau}}_t^{k_a} \|^2 \right], \tag{6}$$

where $\check{\boldsymbol{\tau}}_t^k = \sqrt{\frac{\bar{\alpha}_k}{\bar{\alpha}_{k_a}}} \check{\boldsymbol{\tau}}_t^{k_a} + \sqrt{\frac{\bar{\alpha}_{k_a} - \bar{\alpha}_k}{\bar{\alpha}_{k_a}}} \epsilon$. Once the training converges, we obtain the noise predictor $\epsilon_\theta(\check{\boldsymbol{\tau}}_t, k)$, which achieves the largest SNR at $k = k_a$ as described in Proposition 4.4. It should be noted that for each sample $\check{\boldsymbol{\tau}}_t$, we focus solely on whether the RL component at the center position ($\check{\boldsymbol{z}}_t$) is corrupted, rather than evaluating the entire trajectory slice. For this purpose, we further define $e_\theta(\check{\boldsymbol{z}}_t) = \|\epsilon_\theta(\check{\boldsymbol{\tau}}_t, k_a)_{H+1}\|_F^2$, where $(\cdot)_{H+1}$ represents the (H+1)-th column of the matrix. Subsequently, we utilize $e_\theta(\check{\boldsymbol{z}}_t)$ to evaluate every samples within the partially corrupted dataset, and rescale the prediction range to $[0, 1]$. With a manually defined threshold $\zeta$, samples with $e_\theta(\check{\boldsymbol{z}}_t) > \zeta$ are classified as noised samples ($\mathcal{D}_n$), while the remaining samples are considered clean ($\mathcal{D}_c$).

**Naive DDPM for Corrupted Samples Recovery.** Once the corrupted samples have been identified, the remaining uncorrupted samples can be utilized to train a denoiser (through naive DDPM), which can subsequently be applied to correct the corrupted samples.

To avoid overfitting of DDPM to misclassified noisy data, we reuse $\zeta$ to filter out training data for the naive DDPM. We denote $\mathbb{I}_t$ as a binary indicator variable that specifies whether $\check{\boldsymbol{z}}_t$ is corrupted ($\mathbb{I}_t = 0$ for $e_\theta(\check{\boldsymbol{z}}_t) > \zeta$) or not ($\mathbb{I}_t = 1$ for $e_\theta(\check{\boldsymbol{z}}_t) \leq \zeta$). Given the mask defined as $\boldsymbol{m}_t := [\mathbb{I}_{t-H}, \ldots, \mathbb{I}_{t+H}]$, the training loss for naive DDPM follows

$$\mathbb{E}_{\check{\boldsymbol{\tau}}_t \sim \mathcal{D}_c, k \sim [1, K], \epsilon \sim \mathcal{N}(\mathbf{0}, \boldsymbol{I})} [\|[\epsilon_\phi(\check{\boldsymbol{\tau}}_t^k, k) - \epsilon)] \odot \boldsymbol{m}_t\|^2]. \tag{7}$$

where $\odot$ is the Hadamard product. We refer to this training process as *selective training* in the following discussion. After the training coverges, we then conduct revese DDPM process $p_\phi(\check{\boldsymbol{\tau}}_t^{0:k_a})$ as described in Eq. (2) to all samples within $\mathcal{D}_n$. Finally, we combine the denoised $D_n$ with $D_c$ to form the final dataset. More implementation details of ADG can be found in Appendices B and C.

# 5 Experiments

In this section, we conduct comprehensive experiments to empirically evaluate ADG by exploring three key questions: (1) How does ADG enhance the performance of both non-robust and robust offline RL methods across various data corruption scenarios? (2) What is the individual contribution of ambient loss and selective training to the overall effectiveness of ADG? (3) What are the advantages of ADG's structure, which incorporates two independent diffusion models?

Table 1: Performance under random data corruption. Results are averaged over four random seeds.

| Attack Element | Task | CQL | | IQL | | RIQL | | DT | | RDT | |
|---|---|---|---|---|---|---|---|---|---|---|---|
| | | Naive | ADG | Naive | ADG | Naive | ADG | Naive | ADG | Naive | ADG |
| State | halfcheetah | 15.9±1.8 | 23.9±5.6 | 19.2±2.2 | 28.8±1.7 | 19.9±2.1 | 27.8±4.5 | 27.5±2.5 | 39.8±0.4 | 30.8±1.8 | 34.2±1.4 |
| | hopper | 55.4±6.4 | 78.8±11.1 | 47.6±7.1 | 72.2±7.1 | 34.0±13.4 | 66.3±15.9 | 51.3±14.0 | 79.1±6.7 | 56.6±2.9 | 65.2±7.5 |
| | walker2d | 39.9±5.2 | 46.0±9.7 | 17.5±6.8 | 27.9±3.7 | 14.2±1.2 | 39.5±4.3 | 47.6±4.9 | 55.1±8.9 | 53.4±4.0 | 66.4±2.9 |
| | halfcheetah(10%) | 11.0±1.1 | 17.3±2.7 | 6.1±1.3 | 12.0±0.9 | 4.4±0.9 | 8.3±2.2 | 6.3±0.4 | 26.4±3.0 | 8.3±1.5 | 22.4±0.8 |
| | hopper(10%) | 1.8±0.7 | 3.1±0.6 | 13.3±3.3 | 19.6±6.3 | 15.5±5.4 | 15.7±6.1 | 36.1±7.6 | 37.4±9.4 | 40.8±3.5 | 42.4±7.9 |
| | walker2d(10%) | -0.0±0.1 | 1.0±1.1 | 10.9±7.2 | 15.7±2.8 | 9.2±4.4 | 10.1±4.3 | 18.0±2.5 | 34.0±5.5 | 20.3±2.8 | 20.5±2.7 |
| | kitchen-complete | 3.8±2.8 | 15.0±6.8 | 33.6±7.3 | 51.2±4.5 | 37.5±6.4 | 52.5±4.3 | 37.0±6.2 | 61.9±12.4 | 52.8±1.8 | 58.1±10.1 |
| | kitchen-partial | 0.0±0.0 | 0.6±1.1 | 13.5±3.4 | 23.8±2.2 | 25.9±3.4 | 26.9±7.6 | 31.0±8.1 | 43.8±1.3 | 36.8±5.8 | 49.4±11.1 |
| | kitchen-mixed | 0.0±0.0 | 0.0±0.0 | 16.2±5.6 | 41.2±4.1 | 21.6±3.7 | 41.9±3.2 | 31.8±3.4 | 36.3±13.1 | 41.8±4.3 | 37.5±17.6 |
| | door(1%) | -0.3±0.0 | -0.4±0.0 | 46.6±17.5 | 63.3±9.8 | 39.0±16.4 | 77.5±10.2 | 94.6±4.2 | 102.9±0.4 | 102.8±2.4 | 103.9±2.2 |
| | hammer(1%) | 0.2±0.0 | 0.2±0.0 | 64.6±17.3 | 78.2±17.3 | 70.0±12.6 | 88.4±5.2 | 97.8±12.3 | 115.7±0.7 | 113.8±1.6 | 126.6±26.9 |
| | relocate(1%) | -0.3±0.1 | -0.3±0.1 | 9.4±3.5 | 14.4±4.9 | 5.2±5.0 | 27.8±4.8 | 61.6±5.6 | 67.6±4.0 | 65.0±6.2 | 67.2±4.8 |
| | **Average Score** | 10.6 | **15.4** | 24.9 | **37.4** | 24.7 | **40.2** | 45.0 | **58.3** | 51.9 | **57.8** |
| | **Improvement ↑** | | **45.28%** | | **50.20%** | | **62.75%** | | **29.56%** | | **11.37%** |
| Full-Elelement | halfcheetah | 0.3±0.2 | 33.9±2.6 | 15.5±1.5 | 37.4±1.6 | 22.5±2.3 | 32.3±3.6 | 28.0±5.0 | 39.2±0.9 | 20.7±2.9 | 38.1±1.2 |
| | hopper | 0.7±0.0 | 20.3±7.3 | 26.5±19.6 | 36.1±20.1 | 16.4±3.2 | 49.4±19.7 | 53.0±14.2 | 55.1±11.9 | 58.6±11.1 | 51.2±20.1 |
| | walker2d | -0.1±0.0 | 22.2±4.2 | 20.3±6.9 | 21.1±8.2 | 17.5±3.6 | 19.9±5.9 | 51.0±14.5 | 55.3±1.9 | 56.5±11.0 | 56.1±15.4 |
| | halfcheetah(10%) | 0.9±0.2 | 20.3±4.0 | 4.2±1.0 | 18.3±2.0 | 2.0±0.5 | 14.6±2.4 | 10.0±4.5 | 28.3±2.6 | 16.8±5.1 | 25.2±2.1 |
| | hopper(10%) | 2.0±1.2 | 10.9±12.1 | 9.9±0.3 | 19.1±10.0 | 11.6±2.4 | 14.2±4.8 | 28.3±8.8 | 35.9±6.0 | 36.5±4.7 | 38.4±7.6 |
| | walker2d(10%) | 0.9±1.6 | 1.2±2.0 | 5.4±2.7 | 10.4±2.1 | 4.7±1.2 | 19.7±8.2 | 18.9±4.0 | 32.0±16.6 | 24.5±5.9 | 43.4±9.3 |
| | kitchen-complete | 4.4±3.2 | 13.1±5.4 | 23.8±6.0 | 48.8±3.8 | 26.±5.0 | 48.8±4.5 | 51.3±9.4 | 55.0±5.3 | 60.6±5.7 | 60.0±4.7 |
| | kitchen-partial | 1.9±2.1 | 0.0±0.0 | 1.1±1.4 | 27.5±6.8 | 0.5±0.9 | 15.6±8.7 | 34.4±7.4 | 37.5±7.3 | 45.6±23.7 | 43.8±9.4 |
| | kitchen-mixed | 0.0±0.0 | 1.2±2.2 | 0.8±0.8 | 37.5±5.0 | 8.4±1.8 | 43.1±6.2 | 21.9±13.3 | 23.8±22.4 | 39.4±17.1 | 35.6±9.7 |
| | door(1%) | -0.3±0.0 | -0.3±0.0 | 47.1±12.7 | 56.3±13.9 | 64.3±11.1 | 71.1±0.8 | 33.9±24.3 | 45.1±26.3 | 92.1±17.5 | 105.6±2.0 |
| | hammer(1%) | 0.2±0.0 | 0.2±0.0 | 65.7±12.7 | 84.5±12.2 | 88.7±18.0 | 95.0±28.3 | 26.3±16.9 | 43.3±38.4 | 113.1±23.8 | 109.2±17.3 |
| | relocate(1%) | -0.3±0.0 | -0.3±0.0 | 7.2±2.0 | 10.6±1.7 | 12.5±1.5 | 20.3±8.2 | 35.9±18.7 | 42.0±21.1 | 56.7±13.2 | 61.8±3.4 |
| | **Average Score** | 0.9 | **10.2** | 19.0 | **34.0** | 22.9 | **37.0** | 32.7 | **41.0** | 51.8 | **55.7** |
| | **Improvement ↑** | | **1033.33%** | | **78.95%** | | **61.57%** | | **25.38%** | | **7.52%** |

## 5.1 Experimental Setups

We assess ADG on various widely used offline RL benchmarks [9], including MuJoCo, Kitchen, and Adroit. Since prior work [34] has shown that the impact of data corruption becomes more severe when data is limited, we further evaluate the effectiveness of ADG across different dataset scales by conducting experiments on down-sampled tasks. Specifically, we down-sample 10% and 1% of the data from MuJoCo and Adroit tasks as the new testbed. We also include results under different down-sample ratio of 2% and 5% in Appendix D.7. For MuJoCo, we select "medium-replay" datasets. Additional results on "medium" and "expert" datasets are deferred to Appendix D.4.

For data corruption, we consider two data corruption scenarios: *random* and *adversarial* corruption, as introduced in Section 3. These scenarios are applied either to states alone or to state-action-reward triplets (full-element). Following the settings of prior works [37], we set the corruption rate $\eta$ to 0.3 and the corruption scale $\alpha$ to 1.0. We provide the further implementation details on data corruption in Appendix C.1. We also investigate ADG under Gaussian noise corruption and under different noise ratio and scales in Appendix D.3, D.8.

We evaluate a diverse set of offline RL methods, including pessimistic value estimation method CQL [19], policy constraint methods along with their robust variants IQL [18] and RIQL [37], as well as sequence modeling methods such as DT [6] and RDT [34]. To ensure the robustness of the findings, each experiment is conducted using four distinct random seeds, with the standard deviation across these seeds also reported.

## 5.2 Evaluation under Different Data Corruption

**Results under Random Corruption.** We evaluate the improvement brought by ADG on various offline RL algorithms under random data corruption. The average scores presented in Table 1 show that ADG provides benefits to both non-robust and robust algorithms across all scenarios. Notably, ADG brings significant improvements to MDP-based algorithms (CQL, IQL, and RIQL), with an

average performance boost of 69.1%, demonstrating the effectiveness of sequence completion for missing information. ADG also provides substantial improvements to non-MDP algorithms (DT and RDT), with an average performance boost of 17.4%. When equipped with ADG, both IQL and DT outperform their *Naive* robust variants in nearly all scenarios. This highlights ADG's core strength: by modifying only the dataset, it enables standard algorithms to surpass their robust variants explicitly designed for noise resistance. Moreover, this advantage can be further amplified using robust variants, achieving nearly a 60% improvement on RIQL. Additionally, when equipped with ADG, most algorithms achieve similar performance under both state and full-element corruption, further demonstrating the scalability of ADG. We provide visualizations of ADG's detection and denoising in Appendices D.1 and D.2 to clearly explain its effectiveness, and investigate the effect of using ADG solely for filtering in Appendix D.5.

Table 2: Performance under adversarial data corruption. Results are averaged over four random seeds.

| Attack | Task | CQL Naive | CQL ADG | IQL Naive | IQL ADG | RIQL Naive | RIQL ADG | DT Naive | DT ADG | RDT Naive | RDT ADG |
|---|---|---|---|---|---|---|---|---|---|---|---|
| State | halfcheetah | 31.2±1.4 | 35.6±1.3 | 23.1±6.7 | 29.4±2.2 | 26.5±5.3 | 27.4±1.9 | 34.8±0.6 | 35.6±2.1 | 34.5±1.9 | 35.2±1.3 |
| | hopper | 55.3±11.3 | 57.2±3.2 | 51.0±6.9 | 51.2±26.7 | 41.2±22.7 | 61.1±18.1 | 27.2±10.1 | 55.8±21.2 | 55.2±28.3 | 70.5±7.8 |
| | walker2d | 44.2±6.6 | 43.3±1.4 | 36.2±14.0 | 42.4±7.4 | 16.7±3.7 | 32.4±15.4 | 44.1±8.1 | 62.2±3.2 | 42.4±6.4 | 62.2±5.7 |
| | halfcheetah(10%) | 10.0±2.0 | 18.3±1.8 | 5.6±1.7 | 13.0±4.4 | 3.6±0.5 | 8.4±1.0 | 7.4±0.6 | 26.4±2.1 | 7.5±0.4 | 22.3±1.8 |
| | hopper(10%) | 2.0±1.1 | 2.4±2.8 | 20.0±3.8 | 19.9±2.3 | 19.1±6.9 | 22.6±12.3 | 38.6±4.7 | 40.3±5.9 | 39.3±5.1 | 42.2±6.9 |
| | walker2d(10%) | -0.2±0.1 | 0.2±0.5 | 9.8±3.3 | 14.0±4.3 | 10.8±4.4 | 10.4±4.2 | 22.3±2.4 | 48.4±13.4 | 21.1±2.6 | 41.5±4.8 |
| | kitchen-complete | 3.8±2.8 | 7.5±7.3 | 45.6±2.1 | 53.8±5.7 | 51.9±3.3 | 51.2±2.8 | 48.4±6.7 | 73.8±3.8 | 58.4±3.7 | 60.0±3.1 |
| | kitchen-partial | 0.0±0.0 | 2.5±4.3 | 26.9±7.6 | 33.1±13.5 | 35.4±5.8 | 44.4±8.7 | 32.6±6.1 | 39.4±21.9 | 36.5±8.8 | 42.5±10.5 |
| | kitchen-mixed | 0.0±0.0 | 1.9±3.2 | 41.2±4.5 | 42.5±5.6 | 33.9±11.2 | 38.8±4.5 | 28.2±9.9 | 38.8±9.6 | 30.0±5.5 | 45.6±10.1 |
| | door(1%) | -0.3±0.0 | -0.3±0.0 | 49.3±11.4 | 58.1±10.3 | 47.3±24.8 | 61.6±11.2 | 99.0±0.9 | 103.3±2.0 | 104.7±0.5 | 105.5±1.3 |
| | hammer(1%) | 0.2±0.0 | 0.2±0.0 | 70.4±15.4 | 78.4±8.9 | 69.1±23.4 | 94.1±11.7 | 96.0±2.5 | 120.3±4.3 | 116.6±7.4 | 93.4±13.1 |
| | relocate(1%) | -0.2±0.0 | -0.2±0.1 | 7.0±1.4 | 18.7±8.7 | 17.1±9.5 | 20.5±9.0 | 76.2±5.0 | 80.8±5.9 | 69.0±4.4 | 70.9±8.8 |
| | **Average Score** | 12.2 | **14.0** | 32.2 | **37.9** | 31.1 | **39.4** | 46.2 | **60.4** | 51.3 | **57.6** |
| | Improvement ↑ | | 14.75% | | 17.70% | | 26.69% | | 30.74% | | 12.28% |
| Full-Eelement | halfcheetah | 0.2±0.7 | 30.9±1.6 | 29.3±3.0 | 38.4±1.5 | 17.0±5.9 | 34.5±4.1 | 37.0±2.0 | 39.5±0.5 | 36.3±1.2 | 39.1±0.3 |
| | hopper | 1.1±0.4 | 12.3±1.3 | 51.4±24.1 | 68.8±7.7 | 27.8±12.5 | 40.0±0.8 | 56.4±11.3 | 65.9±18.5 | 62.3±4.6 | 64.7±10.5 |
| | walker2d | -0.3±0.1 | 37.6±13.0 | 42.2±14.3 | 52.7±1.8 | 49.3±6.5 | 57.4±14.7 | 52.4±6.7 | 60.1±7.4 | 58.8±6.0 | 65.2±6.0 |
| | halfcheetah(10%) | -1.1±0.6 | 13.1±1.9 | 4.6±1.2 | 17.1±3.4 | 4.8±2.1 | 17.3±3.7 | 12.7±2.4 | 24.4±1.7 | 14.2±1.5 | 26.7±1.5 |
| | hopper(10%) | 1.0±0.4 | 3.8±2.4 | 21.3±5.4 | 20.5±2.7 | 30.3±5.3 | 35.1±2.4 | 40.0±12.1 | 40.2±3.8 | 39.7±3.8 | 49.4±13.2 |
| | walker2d(10%) | -0.2±0.0 | 0.7±1.1 | 10.8±1.8 | 20.7±4.9 | 14.7±2.7 | 24.6±9.2 | 28.5±7.4 | 37.4±11.3 | 14.0±7.3 | 38.0±10.7 |
| | kitchen-complete | 3.8±2.8 | 7.5±4.7 | 46.2±6.2 | 48.1±6.5 | 50.0±9.8 | 56.2±9.4 | 55.0±6.1 | 65.6±8.0 | 59.4±6.2 | 58.1±13.8 |
| | kitchen-partial | 5.0±4.0 | 4.4±7.6 | 29.4±2.7 | 33.1±6.5 | 19.4±2.1 | 38.1±12.8 | 26.9±17.0 | 42.5±10.2 | 22.5±21.4 | 30.6±12.0 |
| | kitchen-mixed | 0.0±0.0 | 3.8±6.5 | 34.4±4.8 | 37.5±4.7 | 20.6±4.8 | 25.6±3.7 | 38.1±2.1 | 31.9±9.1 | 38.1±15.1 | 43.1±8.2 |
| | door(1%) | -0.3±0.0 | -0.4±0.0 | 66.9±15.0 | 68.1±11.1 | 37.7±3.2 | 55.3±5.7 | 96.5±11.3 | 98.9±3.5 | 87.9±20.5 | 94.1±7.0 |
| | hammer(1%) | 0.1±0.1 | 0.2±0.0 | 61.5±10.1 | 92.6±13.5 | 50.9±20.5 | 80.7±13.2 | 75.1±20.6 | 75.2±25.5 | 62.5±26.5 | 75.4±35.3 |
| | relocate(1%) | -0.2±0.0 | -0.3±0.0 | 5.0±3.4 | 6.0±3.1 | 7.9±3.3 | 7.4±4.1 | 54.5±14.3 | 67.7±5.3 | 4.7±4.9 | 6.8±6.3 |
| | **Average Score** | 0.8 | **9.5** | 33.6 | **42.0** | 27.5 | **39.4** | 47.8 | **54.1** | 41.7 | **49.3** |
| | Improvement ↑ | | 1087.50% | | 25.00% | | 43.27% | | 13.18% | | 18.23% |

**Results under Adversarial Corruption.** We further examine the robustness of ADG under adversarial data corruption. The results, summarized in Table 2, show that ADG consistently improves baseline performance by an average of 24.41%. Notably, when equipped with ADG, the baselines IQL and DT outperform their *Naive* robust variants RIQL and RDT across all scenarios. This further supports the conclusions drawn from the random corruption scenarios. These findings highlight ADG's ability to adapt to and mitigate adversarial data corruption.

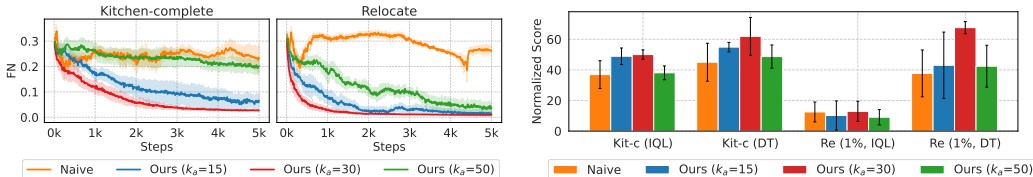

Figure 3: The FN rate during detector training (left), and the performance of IQL and DT using these detectors (right). "Kit-c" denotes Kitchen-complete, and "Re" denotes Relocate.

## 5.3 Ablation Study

We conduct ablation studies to analyze the impact of each component on ADG's performance.

**Impact of Ambient Loss.** We assess the impact of ambient loss on detector performance by varying $t_n \in \{15, 30, 50\}$ on the "kitchen-complete-v0" and "relocate-expert-v1" datasets, selected for

their complexity. Following Section 5.1, Random State Attacks are introduced. Samples detected as corrupted are labeled as positive and others as negative. The detector is trained for 5k steps. We detect the dataset with $e_\theta(\check{z}) \leq \zeta$ (as in Section 4.4) at each step and plot false negatives (FNs), which represent the proportion of undetected corrupted samples. We also evaluate the D4RL scores of baseline algorithms on the datasets recovered using ADG with these trained detectors. As shown in Figure 3, the detector trained with the naive diffusion loss shows some detection capability initially, but quickly overfits to the corrupted portion of the training data. Ambient loss significantly improves it. Notably, $t_n = 30$, which is also used in the main experiments, achieves the lowest false negatives (FNs), implying that nearly all selected samples remain unaltered by attacks. This ensures that the denoiser is trained on nearly clean data, making the naive diffusion loss feasible. Moreover, the D4RL scores of baseline algorithms exhibit a clear correlation with detection performance.

**Impact of Selective Training.** To evaluate the impact of selective training on denoiser, we vary $\zeta$ within the range $\{0.05, 0.10, 0.20, 0.50\}$ and measure the mean squared error (MSE) between the recovered dataset and the ground truth. The results are shown in Figure 4. A very low $\zeta = 0.05$ leads to poor denoiser performance and eventual overfitting, likely due to the insufficient information in a small dataset. Therefore, a very high $\zeta = 0.50$ also degrades performance, possibly due to excessive corrupted data in the dataset. Moderate values of $\zeta = 0.10$ or $0.20$ yield similar, good performance, supporting the necessity of selective training and suggesting that $\zeta$ is somewhat robust, performing well within a certain range. The D4RL scores of the baseline algorithms exhibit a strong correlation with the performance of the denoiser, further supporting the necessity of using $\zeta$ for the selective training of the denoiser.

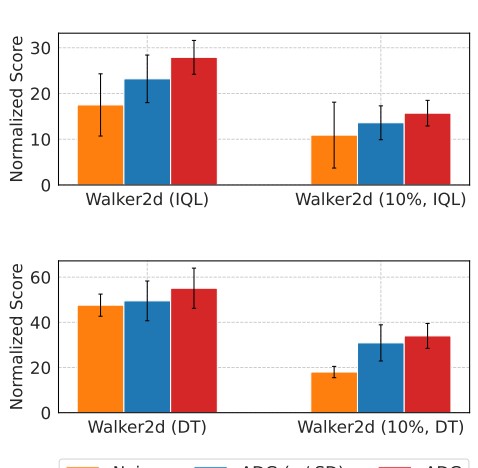

Figure 4: Results on "walker2d-medium-replay-v2" with Random State Attack (Appendix C.1): (a) MSE vs. ground truth, and (b) DT performance across $\zeta$. Best results at $\zeta = 0.20$.

**Impact of Using Two Separate Diffusion Models for Detection and Denoising.** As described in Section 4.4, our approach employs a structure with two independent diffusion models: one serving as the detector and the other as the denoiser. However, it is technically possible to employ a single diffusion model for both detection and denoising tasks by training it concurrently with both Eq. 4 and Eq. 7. We conduct ablation experiments on the "walker2d-medium-replay-v2" dataset under Random State Attack, using both full and limited dataset sizes to evaluate this configuration. The results are presented in Figure 5. Although ADG with single diffusion consistently improves the performance of all baselines, ADG with two independent diffusion models always outperforms ADG with a single model. This result is intuitive, as applying two different losses to a single diffusion model can easily cause interference, leading the network to converge to local minima. Using two independent models effectively addresses this issue by decoupling the mutual interference. For more implementation details and results, refer to Appendix D.10.

Figure 5: Comparison results among Naive, ADG with a single diffusion model, and ADG with separate diffusion models for IQL (upper) and DT (lower). ADG (w/ SD) denotes ADG using a single diffusion model.

Additionally, we also include the ablation study on the length of the slice window by varying the values of $H$ in Appendix D.6. The performance of ADG shows a positive correlation with the hyperparameter $H$ as it increases from 0, and becomes

robust to further changes once $H$ reaches a certain range. This demonstrates the importance of incorporating sequential information and the robustness of ADG.

# 6 Conclusion

We propose Ambient Diffusion-Guided Dataset Recovery (ADG), the first diffusion-based denoising framework for offline RL under data corruption during the training process. We introduce Ambient Denoising Diffusion Probabilistic Models (DDPM), which enable the diffusion model to distinguish between corrupted and clean samples. This mechanism effectively filters the training data, allowing training a naive diffusion model to serve as a denoiser. Comprehensive empirical studies on D4RL benchmarks demonstrate that ADG consistently improves the performance of existing offline RL algorithms across various types, scales, and ratios of data corruption, and in most cases, allows baseline algorithms to outperform their robust variants. We hope that this work establishes a new paradigm for more robust learning from noisy or corrupted data, ultimately benefiting the application of offline RL in real-world scenarios.

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

# A Theoretical Interpretations

## A.1 Proof for Corollary 4.1

Firstly, we define $x^{k_a} = \sqrt{\bar{\alpha}_{k_a}}x^0 + \sqrt{1 - \bar{\alpha}_{k_a}}\epsilon_1$ and $x^k = \sqrt{\bar{\alpha}_k}x^0 + \sqrt{1 - \bar{\alpha}_k}\epsilon_2$, where $\epsilon_1, \epsilon_2 \sim \mathcal{N}(0, I)$. According to the forward process in DDPM, $x^k$ can be expressed as a function of $x^{k_a}$:

$$
\begin{aligned}
x^k &= \sqrt{\bar{\alpha}_k}x^0 + \sqrt{1 - \bar{\alpha}_k}\epsilon_2 \\
&= \sqrt{\bar{\alpha}_k}\frac{x^{k_a} - \sqrt{1 - \bar{\alpha}_{k_a}}\epsilon_1}{\sqrt{\bar{\alpha}_{k_a}}} + \sqrt{1 - \bar{\alpha}_k}\epsilon_2 \\
&= \frac{\sqrt{\bar{\alpha}_k}}{\sqrt{\bar{\alpha}_{k_a}}}x^{k_a} + \sqrt{\sqrt{1 - \bar{\alpha}_{k_a}}^2 - \frac{\sqrt{\bar{\alpha}_k}\sqrt{1 - \bar{\alpha}_{k_a}}}{\sqrt{\bar{\alpha}_{k_a}}}^2}\epsilon \\
&= \sqrt{\frac{\bar{\alpha}_k}{\bar{\alpha}_{k_a}}}x^{k_a} + \sqrt{\frac{\bar{\alpha}_{k_a} - \bar{\alpha}_k}{\bar{\alpha}_{k_a}}}\epsilon
\end{aligned}
\tag{8}
$$

Following the proof sketch of Lemma A.4 in [8], we apply Tweedie's formula to the pair of $x^k$ and $x^0$:

$$
\nabla \log p_k(x^k) = \frac{\sqrt{\bar{\alpha}_k}\mathbb{E}[x^0|x^k] - x^k}{1 - \bar{\alpha}_k}.
\tag{9}
$$

Similarly, applying Tweedie's formula to the pair of $x^k$ and $x^{k_a}$, we derive:

$$
\nabla \log p_k(x^k) = \frac{\sqrt{\frac{\bar{\alpha}_k}{\bar{\alpha}_{k_a}}}\mathbb{E}[x^{k_a}|x^k] - x^k}{\frac{\bar{\alpha}_{k_a} - \bar{\alpha}_k}{\bar{\alpha}_{k_a}}}
\tag{10}
$$

Equating Eq. (9) and Eq. (10), we obtain:

$$
\mathbb{E}[x^{k_a}|x^k] = \frac{\bar{\alpha}_{k_a} - \bar{\alpha}_k}{\sqrt{\bar{\alpha}_{k_a}} \cdot (1 - \bar{\alpha}_k)}\mathbb{E}[x^0|x^k] + \frac{\sqrt{\bar{\alpha}_{k_a}} \cdot (1 - \bar{\alpha}_{k_a})}{\sqrt{\bar{\alpha}_k} \cdot (1 - \bar{\alpha}_k)}x^k
\tag{11}
$$

Since we aim to predict the Gaussian noise rather than the original signal, the forward process of DDPM provides the relationship:

$$
\mathbb{E}[x^0|x^k] = \frac{x^k - \sqrt{1 - \bar{\alpha}_k}\mathbb{E}[\epsilon|x^k]}{\sqrt{\bar{\alpha}_k}}
\tag{12}
$$

Substituting Eq. (12) into Eq. (11), we relate $\mathbb{E}[\epsilon|x^k]$ to $\mathbb{E}[x^{k_a}|x^k]$:

$$
\mathbb{E}[x^{k_a}|x^k] = \frac{\bar{\alpha}_{k_a}}{\sqrt{\bar{\alpha}_k \cdot \bar{\alpha}_{k_a}}}x^k - \frac{(\bar{\alpha}_{k_a} - \bar{\alpha}_k)\mathbb{E}[\epsilon|x^k]}{\sqrt{\bar{\alpha}_{k_a} \cdot \bar{\alpha}_k} \cdot (1 - \bar{\alpha}_k)}
\tag{13}
$$

Building on Theorem A.3 in [8], we minimize the objective:

$$
\mathbb{E}_{x^{k_a}}\mathbb{E}_{k\sim\mathcal{U}(k_a,K)}\mathbb{E}_{x^k|x^{k_0}}\left[\|g_\theta(x^k, k) - x^{k_a}\|^2\right],
\tag{14}
$$

where the minimizer satisfies $g_{\theta^*}(x^k, k) = \mathbb{E}[x^{k_a}|x^k]$. Substituting Eq. (13) into Eq. (14), we have the final form of the optimization objective:

$$
\mathbb{E}_{x^{k_a}}\mathbb{E}_{k\sim\mathcal{U}(k_a,K)}\mathbb{E}_{x^k|x^{k_0}}\left[\|\frac{\bar{\alpha}_{k_a}}{\sqrt{\bar{\alpha}_k \cdot \bar{\alpha}_{k_a}}}x^k - \frac{(\bar{\alpha}_{k_a} - \bar{\alpha}_k)\epsilon_\theta(x^k, k)}{\sqrt{\bar{\alpha}_{k_a} \cdot \bar{\alpha}_k} \cdot (1 - \bar{\alpha}_k)} - x^{k_a}\|^2\right]
\tag{15}
$$

with the minimizer $\epsilon_{\theta^*}(x^k, k) = \mathbb{E}[\epsilon|x^k], \forall k \geq k_a$. This completes the proof.

## A.2 Proof for Theorem 4.3

Let $\check{x}$ be the samples that may or may not contain scaled Gaussian noise $\iota \cdot \epsilon$. It can be expressed in the form of:

$$
\check{x} = x^0 + \iota \cdot \mathbb{I}_{\text{noise}} \cdot \epsilon,
\tag{16}
$$

where $x^0 \in \mathbb{R}^{m \times n}$ is the noise-free matrix, $\iota$ is the noise scale, $\epsilon \in \mathbb{R}^{m \times n}$ is the matrix of Gaussian noise with i.i.d. entries $\epsilon_{ij} \sim \mathcal{N}(0, 1)$, and $\mathbb{I}_{\text{noise}}$ is an unknown indicator variable that determines whether noise is present. Accordingly, we define the samples that contain consistent Gaussian noise $\iota \cdot \epsilon$, which is expressed as:

$$
\tilde{x} = x^0 + \iota \cdot \epsilon.
\tag{17}
$$

**Lemma A.1.** *Given $\check{x}$ and $\tilde{x}$ as defined in Eqs. (16) and (17) respectively, let $q(\cdot^k|\cdot^0)$ denote the distribution of DDPM forward process at diffusion timestep $k$, the following inequality always holds:*

$$\mathbb{D}_{KL}[q(\boldsymbol{x}^k|\boldsymbol{x}^0)\|q(\check{\boldsymbol{x}}^k|\check{\boldsymbol{x}}^0)] \leq \mathbb{D}_{KL}[q(\boldsymbol{x}^k|\boldsymbol{x}^0)\|q(\tilde{\boldsymbol{x}}^k|\tilde{\boldsymbol{x}}^0)]. \tag{18}$$

*Proof.* We first denote the probability density function for distirbution $q(\boldsymbol{x}^k|\boldsymbol{x}^0)$, $q(\check{\boldsymbol{x}}^k|\check{\boldsymbol{x}}^0)$, and $q(\tilde{\boldsymbol{x}}^k|\tilde{\boldsymbol{x}}^0)$ as $\Pr(\cdot)$, $\check{\Pr}(\cdot)$, and $\tilde{\Pr}(\cdot)$ respectively. According to the definition of KL divergence, we have:

$$\begin{aligned}
&\mathbb{D}_{KL}[q(\boldsymbol{x}^k|\boldsymbol{x}^0)\|q(\tilde{\boldsymbol{x}}^k|\tilde{\boldsymbol{x}}^0)] - \mathbb{D}_{KL}[q(\boldsymbol{x}^k|\boldsymbol{x}^0)\|q(\check{\boldsymbol{x}}^k|\check{\boldsymbol{x}}^0)] \\
&= \int_{\boldsymbol{x}} \Pr(\boldsymbol{x}) \log \frac{\Pr(\boldsymbol{x})}{\tilde{\Pr}(\boldsymbol{x})} - \int_{\boldsymbol{x}} \Pr(\boldsymbol{x}) \log \frac{\Pr(\boldsymbol{x})}{\check{\Pr}(\boldsymbol{x})} \\
&= \int_{\boldsymbol{x}} \Pr(\boldsymbol{x}) \log \frac{\check{\Pr}(\boldsymbol{x})}{\tilde{\Pr}(\boldsymbol{x})}.
\end{aligned} \tag{19}$$

Note that the probability density function for $q(\check{\boldsymbol{x}}^k|\check{\boldsymbol{x}}^0)$ can be expressed as a mixture of two components: $\check{\Pr}(\boldsymbol{x}) = \gamma\tilde{\Pr}(\boldsymbol{x}) + (1-\gamma)\Pr(\boldsymbol{x})$, where $\gamma \in [0,1]$ represents the probability that the indicator variable $\mathbb{I}_{\text{noise}} = 1$. Substituting this relation into Eq. (19), we have:

$$\begin{aligned}
&\mathbb{D}_{KL}[q(\boldsymbol{x}^k|\boldsymbol{x}^0)\|q(\tilde{\boldsymbol{x}}^k|\tilde{\boldsymbol{x}}^0)] - \mathbb{D}_{KL}[q(\boldsymbol{x}^k|\boldsymbol{x}^0)\|q(\check{\boldsymbol{x}}^k|\check{\boldsymbol{x}}^0)] \\
&= \int_{\boldsymbol{x}} \Pr(\boldsymbol{x}) \log \frac{\gamma\tilde{\Pr}(\boldsymbol{x}) + (1-\gamma)\Pr(\boldsymbol{x})}{\tilde{\Pr}(\boldsymbol{x})} \\
&= -\int_{\boldsymbol{x}} \Pr(\boldsymbol{x}) \log \frac{\tilde{\Pr}(\boldsymbol{x})}{\gamma\tilde{\Pr}(\boldsymbol{x}) + (1-\gamma)\Pr(\boldsymbol{x})} \\
&\overset{i}{\geq} -\log \int_{\boldsymbol{x}} \Pr(\boldsymbol{x}) \frac{\tilde{\Pr}(\boldsymbol{x})}{\gamma\tilde{\Pr}(\boldsymbol{x}) + (1-\gamma)\Pr(\boldsymbol{x})} \\
&\overset{ii}{\geq} -\log \int_{\boldsymbol{x}} \Pr(\boldsymbol{x}) \frac{\tilde{\Pr}(\boldsymbol{x})}{\tilde{\Pr}(\mathrm{x})^\gamma \cdot \Pr(\boldsymbol{x})^{(1-\gamma)}} \\
&= -\log \int_{\boldsymbol{x}} \Pr(\boldsymbol{x})^\gamma \cdot \tilde{\Pr}(\boldsymbol{x})^{(1-\gamma)} \\
&\overset{iii}{\geq} -\log \int_{\boldsymbol{x}} \gamma\Pr(\boldsymbol{x}) + (1-\gamma) \cdot \tilde{\Pr}(\boldsymbol{x}) \\
&= \log[\gamma + (1-\gamma)] = 0,
\end{aligned} \tag{20}$$

where the inequality $(i)$ holds due to Jensen's inequality, and inequality $(i, iii)$ follows from weighted Arithmetic-Geometric mean inequality. They all hold with equality if $\gamma = 1$, i.e., the indicator takes $\mathbb{I}_{\text{noise}} = 1$ for all samples.

**Lemma A.2.** *For any $c > 0$, with the bounded Gaussian noise scale $\iota$ and the samples $\tilde{x}$ from Eq. (17), one can always find $k_a$ such that for any $k \geq k_a$, the KL divergence between $q(\boldsymbol{x}^k|\boldsymbol{x}^0)$ and $q(\tilde{\boldsymbol{x}}^k|\tilde{\boldsymbol{x}}^0)$ satisfying:*

$$\mathbb{D}_{KL}[q(\boldsymbol{x}^k|\boldsymbol{x}^0)\|q(\tilde{\boldsymbol{x}}^k|\tilde{\boldsymbol{x}}^0)] < c. \tag{21}$$

*Proof.* According to the definition of DDPM forward process as in Eq. (8), we have

$$\boldsymbol{x}^k|\boldsymbol{x}^0 = \sqrt{\bar{\alpha}_k}\boldsymbol{x}^0 + \sqrt{1-\bar{\alpha}_k}\boldsymbol{\epsilon} \tag{22}$$

$$\begin{aligned}
\tilde{\boldsymbol{x}}^k|\tilde{\boldsymbol{x}}^0 &= \sqrt{\bar{\alpha}_k}\tilde{\boldsymbol{x}}^0 + \sqrt{1-\bar{\alpha}_k}\boldsymbol{\epsilon}_1 \\
&= \sqrt{\bar{\alpha}_k}(\boldsymbol{x}^0 + \iota \cdot \boldsymbol{\epsilon}) + \sqrt{1-\bar{\alpha}_k}\boldsymbol{\epsilon}_2 \\
&= \sqrt{\bar{\alpha}_k}\boldsymbol{x}^0 + \sqrt{1-\bar{\alpha}_k + \iota^2 \cdot \bar{\alpha}_k}\boldsymbol{\epsilon}
\end{aligned} \tag{23}$$

Note that $\boldsymbol{\epsilon} \in \mathbb{R}^{m \times n}$ is the matrix of Gaussian noise with i.i.d. entries $\epsilon_{ij} \sim \mathcal{N}(0,1)$. By flattening the matrix into a vector, we can explicitly express the probability density function of the distribution $q(\boldsymbol{x}^k|\boldsymbol{x}^0)$ as follows:

$$\Pr(\boldsymbol{x}^k|\boldsymbol{x}^0) = \frac{1}{(2\pi)^{mn/2}|\boldsymbol{\Sigma}|^{1/2}} \cdot \exp\left(-\frac{1}{2}(\boldsymbol{x}^k - \sqrt{\bar{\alpha}_k}\boldsymbol{x}^0)^\top \boldsymbol{\Sigma}^{-1}(\boldsymbol{x}^k - \sqrt{\bar{\alpha}_k}\boldsymbol{x}^0)\right) \tag{24}$$

where $\boldsymbol{\Sigma} \in \mathbb{R}^{mn}$ is a diagonal matrix with all diagonal elements equal to $1 - \bar{\alpha}_k$. Similarly, the probability density function of the distribution $q(\tilde{\boldsymbol{x}}^k | \tilde{\boldsymbol{x}}^0)$ follows:

$$\tilde{\Pr}(\tilde{\boldsymbol{x}}^k | \boldsymbol{x}^0) = \frac{1}{(2\pi)^{mn/2} |\tilde{\boldsymbol{\Sigma}}|^{1/2}} \cdot \exp\left(-\frac{1}{2}(\tilde{\boldsymbol{x}}^k - \sqrt{\bar{\alpha}_k}\boldsymbol{x}^0)^\top \tilde{\boldsymbol{\Sigma}}^{-1}(\tilde{\boldsymbol{x}}^k - \sqrt{\bar{\alpha}_k}\boldsymbol{x}^0)\right) \quad (25)$$

where $\tilde{\boldsymbol{\Sigma}} \in \mathbb{R}^{mn}$ is a diagonal matrix with all diagonal elements equal to $1 - \bar{\alpha}_k + \iota^2 \bar{\alpha}_k$. Then we can derive the KL divergence between $q(\boldsymbol{x}^k | \boldsymbol{x}^0)$ and $q(\tilde{\boldsymbol{x}}^k | \tilde{\boldsymbol{x}}^0)$:

$$\begin{aligned}
&\mathbb{D}_{KL}[q(\boldsymbol{x}^k | \boldsymbol{x}^0) \| q(\tilde{\boldsymbol{x}}^k | \tilde{\boldsymbol{x}}^0)] \\
&= \frac{1}{2}\left[\log\frac{|\tilde{\boldsymbol{\Sigma}}|}{|\boldsymbol{\Sigma}|} + tr\{\tilde{\boldsymbol{\Sigma}}^{-1}\boldsymbol{\Sigma}\} - mn\right] \\
&= \frac{mn}{2}\left[\log\frac{1 - \bar{\alpha}_k + \iota^2 \bar{\alpha}_k}{1 - \bar{\alpha}_k} + \frac{1 - \bar{\alpha}_k}{1 - \bar{\alpha}_k + \iota^2 \bar{\alpha}_k} - 1\right]
\end{aligned} \quad (26)$$

Let $f(k) = \frac{1 - \bar{\alpha}_k + \iota^2 \bar{\alpha}_k}{1 - \bar{\alpha}_k}$, where $\bar{\alpha}_k$ is a monotonically decreasing function of $k$ with $\bar{\alpha}_k \in [0, 1)$. It is straightforward to deduce that $f(k)$ is also a monotonically decreasing function of $k$, with $f(k) \in [1, \infty)$ if $\iota$ is bounded. Substituting this expression into Eq. (26), we obtain:

$$\mathbb{D}_{KL}[q(\boldsymbol{x}^k | \boldsymbol{x}^0) \| q(\tilde{\boldsymbol{x}}^k | \tilde{\boldsymbol{x}}^0)] = \frac{mn}{2}\left[\log f(k) + \frac{1}{f(k)} - 1\right], \quad (27)$$

which is a monotonically increasing function of $f(k)$ within the range of $f(k)$. It attains its minimum value of 0 when $f(k) = 1$. Therefore, for any $c > 0$, we can always find $k_a$ such that for any $k \geq k_a$, the inequality in Eq. (21) holds.

By combining Lemma A.1 and Lemma A.2, and under the validity of Assumption 4.2, the proof of Theorem 4.3 is complete.

## A.3 Proof for Proposition 4.4

We begin with the noise prediction at diffusion timestep $k$. If the noise is perfectly predicted, it should follow the form:

$$\boldsymbol{\epsilon}_{\text{pred}} = \frac{\boldsymbol{x}^k - \sqrt{\bar{\alpha}_k}\boldsymbol{x}^0}{\sqrt{1 - \bar{\alpha}_k}} \quad (28)$$

Notably, any noised/un-noised sample can be expressed as $\check{\boldsymbol{x}} = \boldsymbol{x}^0 + \iota \cdot \mathbb{I}_{\text{noise}} \cdot \boldsymbol{\epsilon}$, where $\boldsymbol{x}^0 \in \mathbb{R}^{m \times n}$ is the noise-free matrix, $\iota$ is the noise scale, $\boldsymbol{\epsilon} \in \mathbb{R}^{m \times n}$ is the matrix of Gaussian noise with i.i.d. entries $\epsilon_{ij} \sim \mathcal{N}(0, 1)$, and $\mathbb{I}_{\text{noise}}$ is an unknown indicator variable that determines whether noise is present. To predict the noise within $\check{\boldsymbol{x}}$ using diffusion timestep $k$, the first step is to perform the original information consistency operation:

$$h_k(\check{\boldsymbol{x}}) = \sqrt{\bar{\alpha}_k} \cdot \check{\boldsymbol{x}} = \sqrt{\bar{\alpha}_k} \cdot (\boldsymbol{x}^0 + \iota \cdot \mathbb{I}_{\text{noise}} \cdot \boldsymbol{\epsilon}). \quad (29)$$

Then substituting Eq.(29) into Eq.(28), we obtain:

$$\boldsymbol{\epsilon}_{\text{pred}} = \frac{\sqrt{\bar{\alpha}_k} \cdot \iota \cdot \mathbb{I}_{\text{noise}}}{\sqrt{1 - \bar{\alpha}_k}} \cdot \boldsymbol{\epsilon} \quad (30)$$

Assume we have a noise predictor $\boldsymbol{\epsilon}_\theta(\cdot, k)$ with prediction error $\boldsymbol{\delta}_\theta^k \sim \mathcal{N}(\mathbf{0}, \sigma_k^2 \boldsymbol{I})$. Given a sample $\check{\boldsymbol{x}}$ that may or may not contain noise, we have the following two criteria:

**Case 1: Noise-free data ($\mathbb{I}_{\text{noise}} = 0$)** Under such circumstance, the model's prediction is entirely determined by the prediction error:

$$\boldsymbol{\epsilon}_\theta(\check{\boldsymbol{x}}^k, k) = \boldsymbol{\epsilon}_{\text{pred}} + \boldsymbol{\delta}_\theta^k = \mathbf{0} + \boldsymbol{\delta}_\theta^k, \quad (31)$$

thereby the expected Frobenius norm squared of the prediction is:

$$\begin{aligned}
\mathbb{E}[\|\boldsymbol{\epsilon}_\theta(\check{\boldsymbol{x}}^k, k)\|_F^2] &= \mathbb{E}[\|\boldsymbol{\delta}_\theta^k\|_F^2] \\
&= \mathbb{E}[\sum_i \sum_j (\delta_{\theta, ij}^k)^2] \\
&= m \cdot n \cdot \sigma_k^2
\end{aligned} \quad (32)$$

**Case 2: Noisy data ($\mathbb{I}_{\text{noise}} = 1$)** For noisy data, the model's prediction includes both the actual noise and the prediction error. Specifically, we have:

$$\boldsymbol{\epsilon}_\theta(\check{\boldsymbol{x}}^k, k) = \boldsymbol{\epsilon}_{\text{pred}} + \boldsymbol{\delta}_\theta^k = \frac{\sqrt{\bar{\alpha}_k} \cdot \iota \cdot \mathbb{I}_{\text{noise}}}{\sqrt{1 - \bar{\alpha}_k}} \cdot \boldsymbol{\epsilon} + \boldsymbol{\delta}_\theta^k, \tag{33}$$

The expected squared Frobenius norm of the prediction is then given by:

$$
\begin{aligned}
& \mathbb{E}[\|\boldsymbol{\epsilon}_\theta(\check{\boldsymbol{x}}^k, k)\|_F^2] \\
&= \mathbb{E}[\|\frac{\sqrt{\bar{\alpha}_k} \cdot \iota \cdot \mathbb{I}_{\text{noise}}}{\sqrt{1 - \bar{\alpha}_k}} \cdot \boldsymbol{\epsilon} + \boldsymbol{\delta}_\theta^k\|_F^2] \\
&\overset{i}{=} \mathbb{E}[\|\frac{\sqrt{\bar{\alpha}_k} \cdot \iota \cdot \mathbb{I}_{\text{noise}}}{\sqrt{1 - \bar{\alpha}_k}} \cdot \boldsymbol{\epsilon}\|_F^2] + \mathbb{E}[\|\boldsymbol{\delta}_\theta^k\|_F^2] + \frac{2\sqrt{\bar{\alpha}_k} \cdot \iota \cdot \mathbb{I}_{\text{noise}}}{\sqrt{1 - \bar{\alpha}_k}} \mathbb{E}[\text{Tr}(\boldsymbol{\epsilon}^\top \boldsymbol{\delta}_\theta^k)] \\
&= \frac{\iota^2 \cdot \bar{\alpha}_k}{(1 - \bar{\alpha}_k)} \cdot \mathbb{E}[\|\boldsymbol{\epsilon}\|_F^2] + m \cdot n \cdot \sigma_k^2 + 0 \\
&= \frac{\iota^2 \cdot \bar{\alpha}_k}{(1 - \bar{\alpha}_k)} \cdot m \cdot n + m \cdot n \cdot \sigma_k^2
\end{aligned} \tag{34}
$$

The last term in equality $(i)$ vanishes because $\boldsymbol{\epsilon}$ and $\boldsymbol{\delta}_\theta^k$ are mutually independent. Combining the results of Eq. (32) and Eq. (34), and substituting into the definition of the Signal-to-Noise Ratio (SNR) from Eq. (5), we derive:

$$
\begin{aligned}
\text{SNR}(k) &:= \frac{\mathbb{E}[\|\boldsymbol{\epsilon}_\theta(\check{\boldsymbol{x}}_{\text{ns}}^k, k)\|_F^2] - \mathbb{E}[\|\boldsymbol{\epsilon}_\theta(\check{\boldsymbol{x}}_{\text{nf}}^k, k)\|_F^2]}{\mathbb{E}[\|\boldsymbol{\epsilon}_\theta(\check{\boldsymbol{x}}_{\text{nf}}^k, k)\|_F^2]} \\
&= \frac{\frac{\iota^2 \cdot \bar{\alpha}_k}{(1 - \bar{\alpha}_k)} \cdot m \cdot n + m \cdot n \cdot \sigma_k^2 - m \cdot n \cdot \sigma_k^2}{m \cdot n \cdot \sigma_k^2} \\
&= \frac{\iota^2 \cdot \bar{\alpha}_k}{(1 - \bar{\alpha}_k) \cdot \sigma_k^2},
\end{aligned} \tag{35}
$$

which complete the proof.

# B  Algorithm Pseudocode

We provide the pseudocode of our proposed Ambient Diffusion-Guided Dataset Recovery (ADG) in Algorithm 1 for a comprehensive overview.

# C  Implementation Details

## C.1  Data Corruption Details during Training Phase

We study two types of corruption: random noise and adversarial noise. For each type, we investigate two categories of elements to attack: (1) State corruption, where only the states in a portion of the samples are corrupted. According to Section 3, an original trajectory is defined as $\tau = (s_0, a_0, r_0, \ldots, s_{T-1}, a_{T-1}, r_{T-1})$. In MDP-based methods like IQL and CQL, corrupting a state $s_t$ affects two transitions: $(s_{t-1}, a_{t-1}, r_{t-1}, s_t)$ and $(s_t, a_t, r_t, s_{t+1})$. (2) Full-element corruption, where the states, actions, and rewards $(s_t, a_t, r_t)$ in a portion of the samples are corrupted, introducing a stronger challenge for robust offline RL algorithms.

We consider the tasks including MuJoCo, Kitchen and Adroit [9]. We select the "medium-replay-v2" datasets in the MuJoCo tasks for our main experiments, using both the full datasets and down-sampled versions (reduced to 10% of the original size). For Adroit tasks, we choose "expert-v0" datasets and down-sample them to 1% of their original dataset. We use the full datasets for the tasks in the Kitchen, as their original dataset size is already limited. To control the overall level of corruption within the datasets, we introduce two parameters $\eta$ and $\alpha$. The parameter $\eta$ represents the proportion of corrupted data within a dataset, while $\alpha$ indicates the scale of corruption across each individual dimension. These settings are consistent with prior works [40, 37, 34]. We outline two types of random data corruption as follows:

---

**Algorithm 1** Ambient Diffusion-Guided Dataset Recovery (ADG)

---

**Require:** Offline partially corrupted dataset $D$, initialized noise predictors detector $\epsilon_\theta$ and the denoiser $\epsilon_\phi$.

  **Step 1: Update the detector $\epsilon_\theta$ using Ambient Loss**

  **for** each iteration **do**

    Sample a trajectory mini-batch $B = \{(\check{z}_{t-H}, \ldots, \check{z}_{t+H})\} \sim D$, where $\check{z}$ represents either the state $s$ or the concatenation of $(s, a, r)$ may or may not contain noise

    Sample uniformly distributed diffusion timestep $k \sim \{k_a, \ldots, K\}$

    Sample random Gaussian noise $\epsilon_t^k \sim \mathcal{N}(\mathbf{0}, \boldsymbol{I})$

    Produce noised element through $\tilde{z}_t^k = \sqrt{\bar{\alpha}_k}\check{z}_t + \sqrt{1 - \bar{\alpha}_k}\epsilon_t^k$

    Get trajectory $\check{\boldsymbol{\tau}}_t = [\check{z}_{t-H}, \ldots, \check{z}_{t+H}] \in \mathbb{R}^{M \times (2H+1)}$

    Update the $\epsilon_\theta$ through Eq. 4.

  **end for**

  **Step 2: Update the denoiser $\epsilon_\phi$ using Naive loss**

  **for** each iteration **do**

    Sample a trajectory mini-batch $B = \{(\check{z}_{t-H}, \ldots, \check{z}_{t+H})\} \sim D$, where $\check{z}$ represents either the state $s$ or the concatenation of $(s, a, r)$ may or may not contain noise

    Sample uniformly distributed diffusion timestep $k \sim \{1, \ldots, K\}$

    Sample random Gaussian noise $\epsilon_t^k \sim \mathcal{N}(\mathbf{0}, \boldsymbol{I})$

    Produce noised element through $\tilde{z}_t^k = \sqrt{\bar{\alpha}_k}\check{z}_t + \sqrt{1 - \bar{\alpha}_k}\epsilon_t^k$

    Get trajectory $\check{\boldsymbol{\tau}}_t = [\check{z}_{t-H}, \ldots, \check{z}_{t+H}] \in \mathbb{R}^{M \times (2H+1)}$

    Get $e_\theta(\check{z}_t)$ using detector $\epsilon_\theta(\check{\boldsymbol{\tau}}_t, k)$ with $k = k_a$

    Get mask $\boldsymbol{m}_t := [\mathbb{I}_{t-H}, \ldots, \mathbb{I}_{t+H}]$, where $\mathbb{I}_t = 0$ for $e_\theta(\check{z}) > \zeta$ and $\mathbb{I}_t = 1$ for $e_\theta(\check{z}) \leq \zeta$

    Update the $\epsilon_\phi$ through Eq. 7.

  **end for**

  **Step 3: Detect and Recover the noised dataset**

  **for** each $\check{z}_t$ in the noised dataset with $e_\theta(\check{z}) > \zeta$ **do**

    Get the trajectory $\check{\boldsymbol{\tau}}_t := [\check{z}_{t-H}, \ldots, \check{z}_{t+H}]$

    Recover the trajectory $\check{\boldsymbol{\tau}}_t = \frac{1}{\sqrt{\bar{\alpha}_k}}\left[\check{\boldsymbol{\tau}}_t^k - \sqrt{1 - \bar{\alpha}_k}\epsilon_\theta\left(\check{\boldsymbol{\tau}}_t^k, k\right)\right]$

    Replace $\check{z}_t$ with $(\check{\boldsymbol{\tau}}_t)_{H+1}$, which represents the (H+1)-th column of $\check{\boldsymbol{\tau}}$

  **end for**

---

- **Random State Attack:** We randomly sample $\eta \cdot N \cdot T$ states from all trajectories, where $N$ refers to the number of trajectories and $T$ represents the number of steps in a trajectory. The selected states are then modified as $\hat{s} = s + \lambda \cdot \text{std}(s)$, where $\lambda \sim \text{Uniform}[-\alpha, \alpha]^{d_s}$. Here, $d_s$ represents the dimension of states, and $\text{std}(s)$ is the $d_s$-dimensional standard deviation of all states in the offline dataset. The noise is scaled based on the standard deviation of each dimension and is independently added to each respective dimension.

- **Random Full-element Attack:** We randomly sample $\eta \cdot N \cdot T$ state-action-reward triplets $(s_t, a_t, r_t)$ from all trajectories, and modify the action $\hat{a} = a + \lambda \cdot \text{std}(a)$, $\hat{r} \sim \text{Uniform}[-30 \cdot \alpha, 30 \cdot \alpha]^{d_s}$, where $\lambda \sim \text{Uniform}[-\alpha, \alpha]^{d_a}$, $d_a$ represents the dimension of actions and $\text{std}(a)$ is the $d_a$-dimensional standard deviation of all actions in the offline dataset. The corruption to rewards is multiplied by 30, following the setting in RDT [34], since offline RL algorithms tend to be resilient to small-scale random rewards corruption.

The two types of adversarial data corruption are detailed as follows:

- **Adversarial State Attack:** We first pretrain IQL agents with a Q-function $Q_p$ and policy function $\pi_p$ on clean datasets. Then, we randomly sample $\eta \cdot N \cdot T$ states and modify them as follows. Specifically, we perform the attack by solving for $\hat{s} = \min_{\hat{s} \in \mathbb{B}_d(s, \alpha)} Q_p(\hat{s}, a)$. Here, $\mathbb{B}_d(s, \epsilon) = \{\hat{s} | |\hat{s} - s| \leq \epsilon \cdot \text{std}(s)\}$ regularizes the maximum difference for each state dimension. The optimization is implemented through Projected Gradient Descent, similar to prior works [25, 42, 37, 34]. In this approach, we first initialize a learnable vector $v \in [-\alpha, \alpha]^{d_s}$, and then conduct a 100-step gradient descent with a step size of 0.01 for $\hat{s} = s + v \cdot \text{std}(s)$. After each update, we clip each dimension of $z$ within the range $[-\alpha, \alpha]$.

- **Adversarial Full-element Attack:** We use the pretrained IQL agent with a Q-function $Q_p$ and a policy function $\pi_p$. Then, we randomly sample $\eta \cdot N \cdot T$ state-action-reward triplets $(s_t, a_t, r_t)$, and modify $u = (s, a)$ to $\hat{u} = \min_{\hat{u} \in \mathbb{B}_d(u, \alpha)} Q_p(u)$. Here, $\mathbb{B}_d(u, \alpha) = \{\hat{u} \| \hat{u} - u \| \leq \alpha \cdot \text{std}(u)\}$ regularizes the maximum difference for each dimension of $u$. The optimization is implemented through Projected Gradient Descent, as discussed above. The rewards in these triplets are also modified to: $\hat{r} = -\alpha \cdot r$.

## C.2 ADG Network Structure

Both detector and denoiser in ADG is classical diffusion models [31] with Unet Structure as shown in 6. The diffusion models are unconditional, which do not need to introduce extra knowledge except the noised input, make ADG can be simply applied to new scenarios. For the noised prediction generation, we utilize a 3 Layer MLP with Mish activation. Further details including the hyperparameters refer to Section C.3.

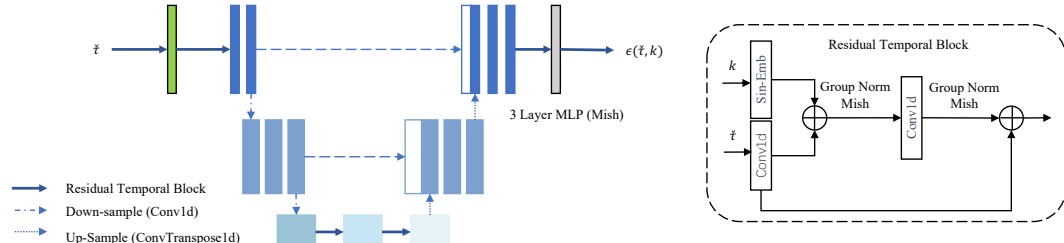

Figure 6: Neural network structure of ADG.

## C.3 Hyperparameters

We present the hyperparameters and other details of ADG in Table 3 and 4.

Table 3: Generic hyperparameters of ADG.

| Hyper-parameter | Value |
|---|---|
| Batch size | 256 |
| Total diffusion step ($K$) | 100 |
| Ambient Nature Timestep ($K_a$) | 30 |
| Threshold $\zeta$ | 0.20 |
| Temporal slice size $H$ | 5 |
| Learning Rate (lr) | 5 |
| Noise prediction network | FC(256,256,256) with Mish activations |
| Dropout for predictor network | 0.1 |
| Variance schedule | Variance Preserving (VP) |
| Learning Rate | 1e-4 |

Table 4: Hyperparameters of ADG for different benchmark environments and datasets.

| Tasks | Embedded Dimension | Denoising Diffusion Steps | Total Training Steps |
|---|---|---|---|
| MuJoCo | 128 | 100 | 20k |
| MuJoCo (10%) | 128 | 100 | 20k |
| Kitchen | 512 | 10 | 40k |
| Adroit (1%) | 512 | 10 | 40k |

## C.4 Dataset Details

Since we conduct the experiments on both the full and down-sampled datasets, we provide detailed information about the number of transitions and trajectories as shown in Table 5.

| Dataset | halfcheetah | hopper | walker2d |
|---|---|---|---|
| # Transitions | 202000 | 402000 | 302000 |
| # Trajectories | 202 | 2041 | 1093 |

| Dataset | halfcheetah(10%) | hopper(10%) | walker2d(10%) |
|---|---|---|---|
| # Transitions | 20000 | 32921 | 27937 |
| # Trajectories | 20 | 204 | 109 |

| Dataset | kitchen-complete | kitchen-partial | kitchen-mixed |
|---|---|---|---|
| # Transitions | 3680 | 136950 | 136950 |
| # Trajectories | 19 | 613 | 613 |

| Dataset | door | hammer | relocate |
|---|---|---|---|
| # Transitions | 10000 | 10000 | 10000 |
| # Trajectories | 50 | 50 | 50 |

Table 5: Detailed information about the number of transitions and trajectories.

## C.5 Computation Overhead

In Table 6, we compare the computational cost of ADG with baseline algorithms on a single GPU (P40). Each algorithm is trained on the "walker2d-medium-replay-v2" dataset, and we record the total training time. CQL requires significantly more time due to its reliance on a larger number of Q ensembles and other computationally intensive processes. IQL and DT exhibit similar computational costs to their robust counterparts, RIQL and RDT. Notably, ADG achieves substantially lower computational costs than all baselines, demonstrating its ability to deliver significant performance gains with minimal additional overhead. The total time and per-step time for training the detector and denoiser, as well as for dataset recovery, are also reported.

Table 6: Training time of ADG and offline RL baseline algorithms.

| Tasks | CQL | IQL | RIQL | DT | RDT | ADG |
|---|---|---|---|---|---|---|
| Epoch Num | 1000 | 1000 | 1000 | 100 | 100 | 10 |
| Total Time (h) | 9.39 | 3.90 | 4.14 | 1.22 | 1.27 | 0.78 |
| Detector Training (h) | | | 0.37 (0.07s/step) | | | |
| Denoiser Training (h) | | | 0.22 (0.04s/step) | | | |
| Dataset Recovering (h) | | | 0.19 | | | |

# D  Additional Experimental Results

We present additional experimental results in this section. The network architecture and hyperparameters remain the same as those specified in Table 3. All results are averaged over four seeds.

## D.1  Visualization of $e(\check{x})$ Predicted by the Detector

To demonstrate the effectiveness of ADG's detection capability, we visualize the squared Frobenius norm $e(\check{x})$ of each sample $\check{x}$ across the MuJoCo, Kitchen, and Adroit datasets using trained detectors. Consistent with the experiments in Section 5, we use "medium-replay" datasets for MuJoCo and "expert" datasets for Adroit. The results are shown in Figure 7 using box plots. From the results, we observe a clear difference between the clean samples and corrupted samples in the distribution of $e(\check{x})$.

## D.2  Visualization of the Recovered Trajectories

To demonstrate the effectiveness of our proposed approach, ADG, we visualize a partial trajectory of "hopper" in Figure 8. From the results, we observe that ADG can effectively detect and recover the

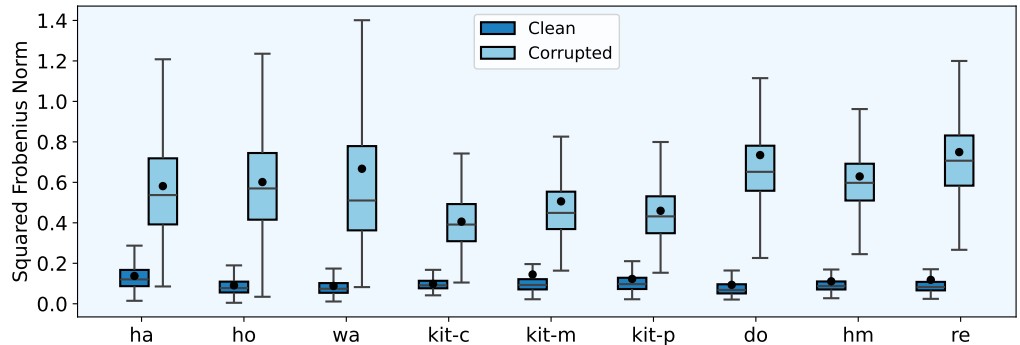

Figure 7: Visualization of the distribution of $e(\check{x})$ for samples in the datasets, including halfcheetah, hopper, walker2d, kitchen-complete, kitchen-mixed, kitchen-partial, door, hammer, and relocate, denoted as "ha", "ho", "wa", "kit-c", "kit-m", "kit-p", "do", "hm", and "re", respectively. We distinguish clean and corrupted samples using different colors.

corrupted samples. The partially noised trajectory has its discontinuity significantly reduced after recovery. Additionally, ADG introduces nearly no further corruption to the clean samples during the recovery process.

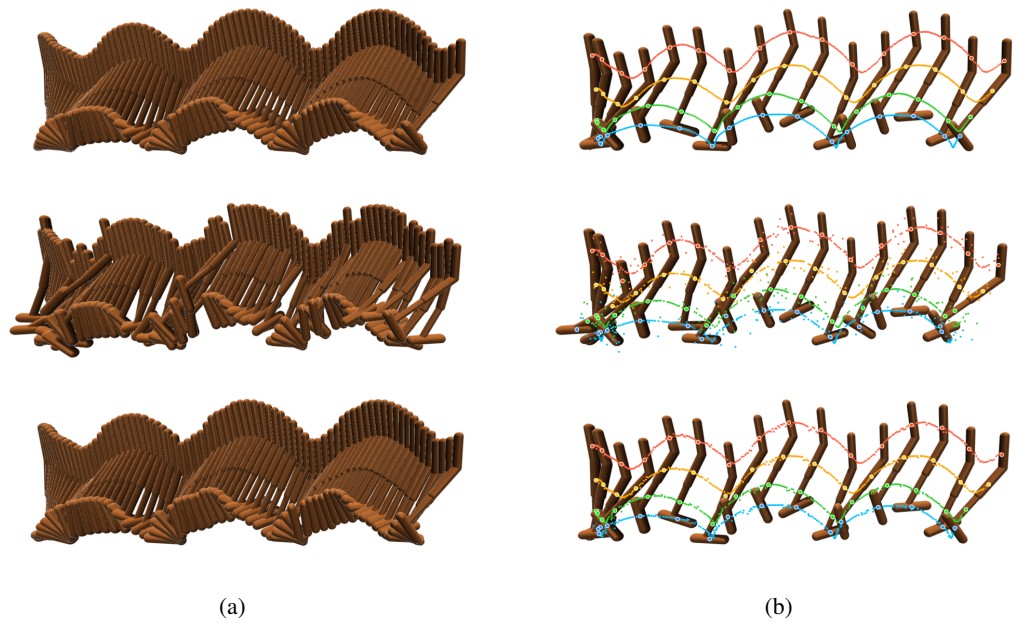

(a)                                                    (b)

Figure 8: Visualization of the denoising effect of ADG on the "hopper-medium-replay-v2" dataset. In (a), we present the complete trajectories, while in (b), we provide a partial view of the trajectories along with their scatter plot representations to highlight the differences in trajectory coherence. In both subfigures, the first row depicts the clean trajectories from the offline dataset, the second row shows the trajectories perturbed with Random State Attack as described in Section C.1, and the third row illustrates the trajectories restored using ADG, which have largely corrected the corrupted samples.

## D.3 Performance Under Gaussian Noise

As presented in Section 5.2, we evaluated the performance of ADG under both random and adversarial noise. We further analyze the robustness of ADG against Gaussian noise in the state.

The implementation simply changes the noise from Uniform $[-\alpha, \alpha]^{d_s}$ in C.1 to Gaussian noise $\mathcal{N}(0, \alpha^2)$. We set $\eta = 0.3$ and $\alpha = 1.0$, as in the other experiments. The results are presented in Table 7, where ADG consistently improves the performance of the baselines, including IQL and DT, further highlighting its efficiency.

Table 7: Results under Gaussian noise.

| Attack | Task | IQL | | DT | |
| --- | --- | --- | --- | --- | --- |
| | | Naive | ADG | Naive | ADG |
| State | halfcheetah | 17.2±4.6 | 27.9±1.3 | 36.1±1.8 | 36.1±1.1 |
| | hopper | 33.3±9.1 | 54.9±2.6 | 41.8±12.6 | 60.8±13.7 |
| | walker2d | 18.0±5.6 | 40.3±16.0 | 50.9±5.2 | 58.2±10.1 |
| | halfcheetah(10%) | 13.6±3.5 | 15.6±3.7 | 12.8±2.1 | 24.8±1.7 |
| | hopper(10%) | 25.4±11.7 | 32.6±4.5 | 39.0±7.4 | 44.3±7.7 |
| | walker2d(10%) | 15.3±5.3 | 6.7±3.1 | 27.1±8.2 | 33.7±4.0 |
| | **Average** **Improvement** ↑ | 20.5 | **29.7** **44.88%** | 34.6 | **43.0** **24.28%** |
| Full-Element | halfcheetah | 10.3±3.9 | 32.5±2.5 | 34.1±2.3 | 44.5±7.3 |
| | hopper | 28.0±9.9 | 31.9±2.6 | 38.1±9.2 | 49.7±9.4 |
| | walker2d | 8.9±2.7 | 38.7±13.4 | 59.3±7.2 | 60.2±3.1 |
| | halfcheetah(10%) | 5.9±1.6 | 17.8±2.8 | 11.8±2.0 | 21.7±2.8 |
| | hopper(10%) | 15.6±3.8 | 13.4±1.7 | 25.5±7.9 | 37.8±12.2 |
| | walker2d(10%) | 11.7±5.7 | 15.8±3.4 | 24.2±6.3 | 27.6±15.8 |
| | **Average** **Improvement** ↑ | 13.4 | **25.0** **86.57%** | 32.2 | **40.2** **24.84%** |

## D.4 Performance Under MuJoCo Dataset with Different Quality Levels

We evaluate the robustness of ADG under varying dataset quality levels. Specifically, we select the "medium-v2" and "expert-v2" datasets from MuJoCo tasks and downsample each to 2% of their original size, resulting in datasets containing 20k samples. For comparison, we include RIQL, DT, and RDT as baselines. We do not include the results of CQL and IQL as they perform poorly on these down-sampled datasets. The results, summarized in Table 8, demonstrate that ADG consistently enhances the performance of these baselines across different dataset quality levels.

Table 8: Results on "medium-v2" and "expert-v2" datasets under Random State Attack.

| Dataset | Task | RIQL | | DT | | RDT | |
| --- | --- | --- | --- | --- | --- | --- | --- |
| | | Naive | ADG | Naive | ADG | Naive | ADG |
| medium | halfcheetah(2%) | 18.0±1.5 | 24.8±2.1 | 15.7±1.3 | 31.1±1.4 | 22.3±1.1 | 22.0±3.0 |
| | hopper(2%) | 47.5±7.3 | 44.1±2.7 | 48.6±3.2 | 51.4±4.0 | 52.2±5.7 | 54.1±4.9 |
| | walker2d(2%) | 25.4±5.0 | 26.4±4.0 | 20.5±6.0 | 50.0±4.7 | 28.0±8.0 | 46.8±3.6 |
| | **Average** **Improvement** ↑ | 30.3 | **31.8** **4.95%** | 28.3 | **44.2** **56.18%** | 34.2 | **41.0** **19.88%** |
| expert | halfcheetah(2%) | 0.5±1.2 | 1.2±0.7 | 2.9±0.8 | 4.2±1.0 | 4.4±0.2 | 2.02±0.6 |
| | hopper(2%) | 32.0±4.4 | 39.7±12.9 | 38.5±7.4 | 41.3±4.3 | 48.6±7.0 | 42.6±7.0 |
| | walker2d(2%) | 21.7±4.6 | 46.6±2.7 | 40.7±4.8 | 53.2±3.2 | 41.6±4.2 | 55.3±6.1 |
| | **Average** **Improvement** ↑ | 18.1 | **29.2** **61.33%** | 27.4 | **32.9** **20.07%** | 31.5 | **33.3** **5.71%** |

## D.5 Filtered Datasets vs. Recovered Datasets in MDP-based Algorithms

For MDP-based algorithms such as IQL and RIQL, which do not take sequences as input, filtering out corrupted samples from the dataset using a detector, without recovery, is also a viable method to

mitigate the impact of corrupted data. Specifically, we construct the filtered dataset by excluding all $\breve{\boldsymbol{x}}$ for which $e_\theta(\breve{\boldsymbol{x}}) > \zeta = 0.20$ using ADG. To further investigate this approach, we also evaluate filtering with $\zeta = 0.10$ to assess whether a stricter threshold improves performance. Although ADG demonstrates strong detection capabilities, some corrupted data may remain in the filtered dataset. Therefore, we also evaluate the performance of the filtered dataset after removing the remaining corrupted data, which we refer to as the purified dataset. We then perform IQL and RIQL on these datasets. The results, shown in Figure 9, reveal the following insights: (1) The filtered dataset does not consistently improve the performance of the MDP-based algorithms. (2) The recovered dataset shows better robustness, even when compared to the purified dataset. (3) Using a lower value of $\zeta$ to filter the dataset more aggressively does not improve performance, possibly due to the reduced information content in the dataset. These findings highlight that dataset size is a crucial factor for RL performance, suggesting that filtering the dataset without recovery does not outperform recovery-based approaches. This emphasizes the importance of recovery in ADG.

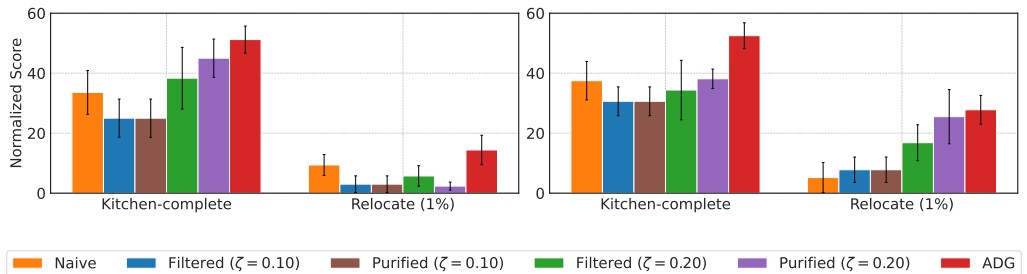

Figure 9: The performance of IQL (left) and RIQL (right) on the noised, filtered, purified, and restored datasets. Performance on the noised data is referred to as "Naive".

### D.6 Evaluation of Performance Across Different Temporal Slice Sizes ($H$)

In this section, we evaluate ADG with varying temporal slice sizes by setting $H \in \{0, 1, 3, 5, 7\}$. As described in Section 4.4, the temporal slice size is defined as $2 \cdot H + 1$. Notably, $H = 0$ means the diffusion model takes only a single step as input and cannot access sequential information. The results are presented in Figure 10. From the results, we observe that the performance of the detector and denoiser in ADG improves significantly as $H$ increases from $0$ to $3$, and remains robust for $H = 3, 5, 7$. The performance of IQL strongly correlates with that of the detector and denoiser, showing significant improvement for $H = 3, 5, 7$ compared to $H = 0, 1$. In contrast, DT is more resilient to variations in ADG's recovery performance across different values of $H$, likely due to its inherent ability to leverage temporal information and mitigate the effects of corruption. This observation aligns with the findings in RDT [34]. These results highlight the critical role of incorporating temporal slices in enhancing the performance of the detector, denoiser, and overall RL performance on the denoised dataset.

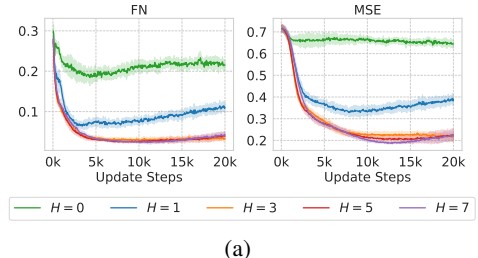
(a)

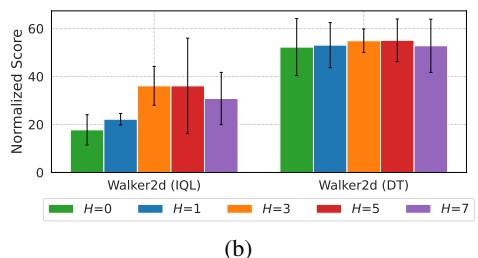
(b)

Figure 10: The performance of the ADG detector and denoiser under varying temporal slice sizes ($H$) is shown in (a), while (b) presents the corresponding performance of the baseline algorithms, IQL and DT, on the "walker2d-medium-replay-v2" dataset. FN represents the False Negative ratio (the proportion of corrupted samples incorrectly identified as clean), and MSE indicates the mean squared error between the restored dataset and the ground-truth dataset.

## D.7 Evaluation under Various Dataset Scales

We further examine the robustness of ADG on MuJoCo tasks with dataset sizes of 20% and 50% of the "medium-replay-v2" datasets. We investigate corruption types including Random State Attack and Random Full-element Attack as described in Section C.1. The comparison results are shown in Figure 11. From the results, we observe that the performance of both the algorithms and their robust variants on both corrupted and restored datasets is positively correlated with dataset size. Moreover, ADG outperforms the baselines across most dataset sizes, validating the effectiveness and importance of dataset recovery.

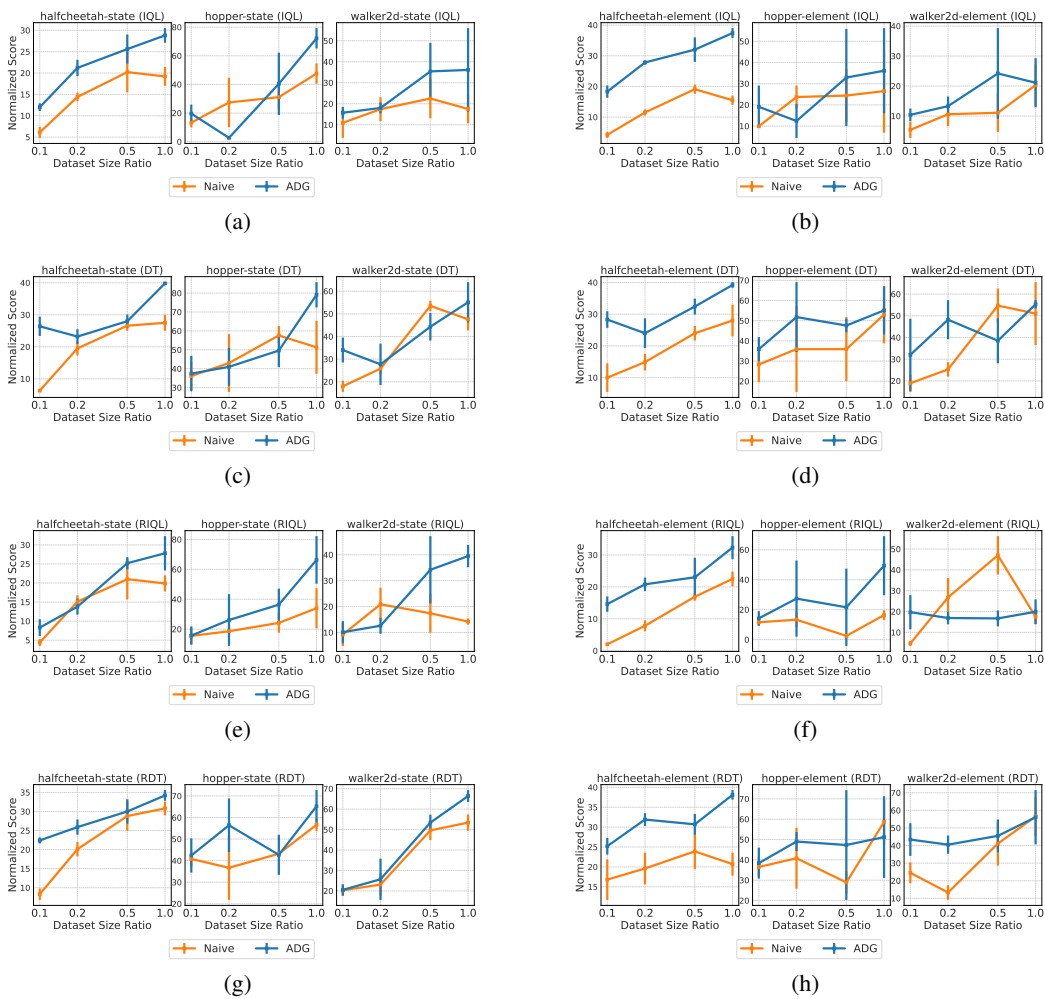

Figure 11: Performance of ADG under random corruption across different dataset scales.

## D.8 Varying Corruption Rates and Scales

We evaluate the robustness of ADG under various corruption rates $\{0.0, 0.1, 0.3, 0.5\}$ and scales $\{0.0, 1.0, 2.0\}$. We choose the "walker2d-medium-replay-v2" dataset downsampled to 10% of its original size to make the results more sensitive to corruption rates and scales. As shown in Figure 12, increasing corruption rates and scales progressively degrade the performance of baseline algorithms like IQL and DT due to greater deviations between the corrupted and clean datasets. Nevertheless, ADG consistently enhances the overall performance of these baselines.

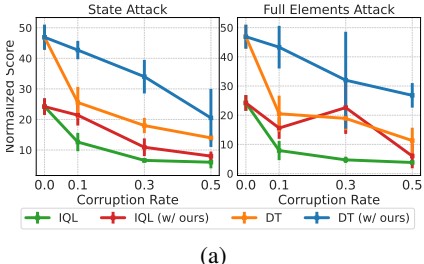 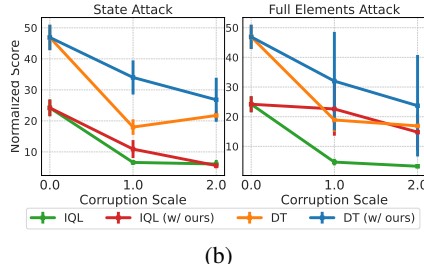

|           (a)           |           (b)           |

Figure 12: Results under various corruption rates (a) and scales (b) on the "walker2d-medium-replay-v2" dataset, which is downsampled to 10% of the original size.

## D.9 Impact of $\zeta$ in Dataset Splitting

As outlined in Section 4.4, we introduce a threshold hyperparameter $\zeta$ to differentiate between noisy and clean samples. To emphasize the importance of selective sampling during training, we evaluate ADG with a trained denoiser by varying $\zeta$ over the range $\{0.02, 0.05, 0.1, 0.2, 0.5, 1.0\}$. Note that when $\zeta = 1$, the dataset remains unchanged. The results are presented in Figure 13. A higher $\zeta$ incorporates more noisy samples into the restored dataset without applying restoration, increasing the risk of overfitting in the naive diffusion model. In contrast, a lower $\zeta$ results in more clean samples being misclassified as noisy, leading to unnecessary restoration and a loss of original dataset information. There is a trade-off in choosing the threshold $\zeta$, and $\zeta = 0.20$ yields the lowest MSE between the restored and clean datasets, as well as the best D4RL score across all baseline algorithms.

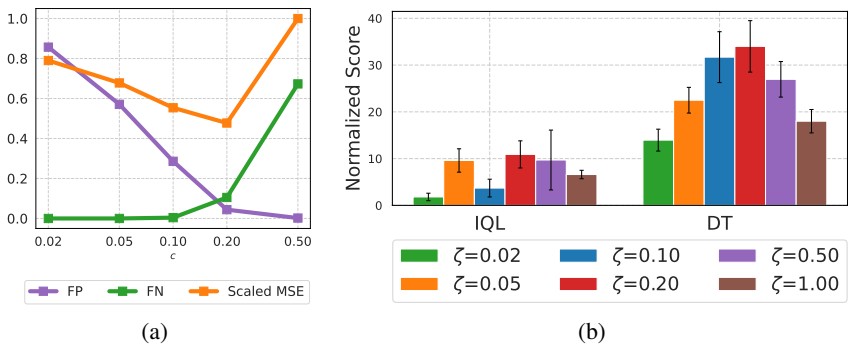

|           (a)           |           (b)           |

Figure 13: Results of (a) the detection performance in the dataset recovery process and (b) D4RL score under various $\zeta$ on the "walker2d-medium-replay-v2" dataset. In (a), we label noisy samples as positive and clean samples as negative. The False Positive (FP) rate represents the ratio of clean samples incorrectly classified as noisy, while the False Negative (FN) rate corresponds to the ratio of noisy samples misclassified as clean. Both are expected to be low for a better dataset splitting result. Scaled MSE is the MSE scaled to $[0, 1]$, which directly reflects the deviation of the restored dataset from the clean dataset.

## D.10 Additional Experiments Comparing the Use of Single vs. Double Diffusion Models

In this section, we provide further details on the implementation of ADG using a single diffusion model. We reorganize Algorithm 1 to merge **Step 1** and **Step 2** into a single loop. In each step, the diffusion model is updated using both Eq. 4 and Eq. 7. We continue to use $k_a$ to detect corrupted data in the partially corrupted dataset. The results for "halfcheetah-medium-replay-v2" and "hopper-medium-replay-v2" datasets are presented in Figure 14 to complete the analysis. Employing a single diffusion model improves performance in 4 out of 6 tasks compared to the baselines. However, ADG with two independent models still achieves significantly higher performance. These findings align with the conclusions in Section 5.3.

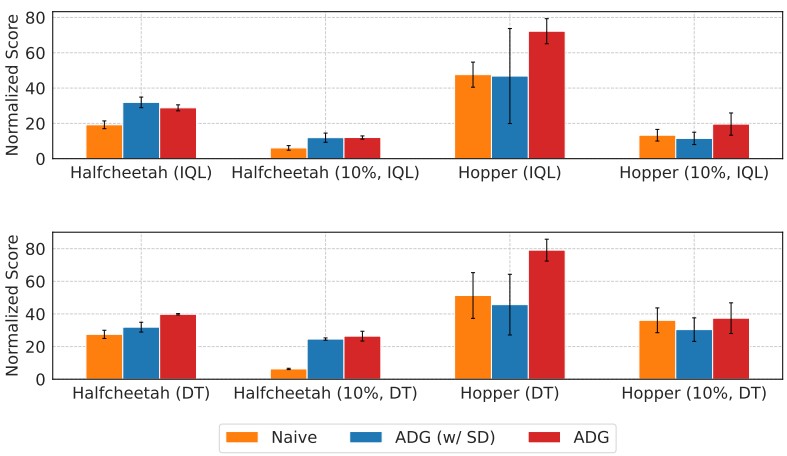

Figure 14: Comparison results among Naive, ADG with a single diffusion model, and ADG with separate diffusion models for IQL (upper) and DT (lower) under "halfcheetah-medium-replay-v2" and "hopper-medium-replay-v2" datasets. ADG (w/ SD) denotes ADG using a single diffusion model.

## D.11   Missing data

While our main focus is on additive corruption, ADG is theoretically applicable to a broader class of noise, including missing data such as dropped elements. We construct a missing-data variant of each dataset by randomly zeroing out 30% of the actions. We evaluate baseline offline RL algorithms and ADG under this setting and report the results averaged over four seeds in Table 9. Despite the substantial information loss, ADG consistently improves performance across all environments, recovering a significant portion of the original policy quality.

Table 9: Performance under missing data. Results are averaged over 4 random seeds. "Missing Data" refers to standard offline RL applied to the corrupted dataset. "Missing Data (w/ ADG)" denotes applying ADG for recovery.

| Environment | Missing Data | Missing Data (w/ ADG) | Clean |
|---|---|---|---|
| halfcheetah-mr | 3.8±0.2 | 31.3±0.7 | 38.9±0.5 |
| hopper-mr | 14.0±1.0 | 45.5±16.4 | 81.8±6.9 |
| walker2d-mr | 7.2±0.3 | 32.7±4.1 | 59.9±2.7 |

These results indicate that ADG is not limited to additive noise settings and generalizes well to missing data, further demonstrating its robustness in practical offline RL scenarios where various forms of corruption may co-occur.

## D.12   Extending ADG to Recent Offline RL Methods

To evaluate the broader applicability of ADG beyond standard baselines, we further apply it to two recent state-of-the-art offline RL algorithms: A2PR [24] and NUNO [17]. Specifically, we test their robustness under the Random State Attack corruption setting and assess the performance gains when combined with ADG. This allows us to examine whether ADG can consistently enhance policy quality across diverse algorithmic backbones.

As shown in Table 10, ADG significantly improves performance for both A2PR and NUNO in all environments, highlighting its general utility in mitigating data corruption across a range of policy learning strategies.

Table 10: Performance of A2PR and NUNO under Random State Attack, with and without ADG. Results are averaged over four random seeds.

| Environment | A2PR (noised) | NUNO (noised) | A2PR (w/ ADG) | NUNO (w/ ADG) |
|---|---|---|---|---|
| halfcheetah-mr | 4.4±0.5 | 27.3±7.8 | 13.2±1.6 | **29.2**±7.5 |
| hopper-mr | 16.1±5.0 | 12.6±0.8 | 25.8±6.2 | **29.9**±0.6 |
| walker2d-mr | 4.6±2.3 | 11.0±5.5 | 9.3±0.8 | **14.6**±4.1 |

## D.13 Recovery under Fully Corrupted Trajectories

In ADG, each trajectory is processed through overlapping temporal slices, where the detector focuses on verifying whether the central element within each slice is corrupted. This design is not a limitation but a deliberate choice: it allows ADG to (1) leverage temporal context from both past and future elements for detection, (2) avoid compounding errors when modeling long trajectories, and (3) ensure a fair comparison with prior methods such as RDT [34], which adopt the same middle-step corruption setup.

Importantly, because slices are generated in a sliding-window manner, every element in the dataset is examined as the center of some slice, meaning that all time steps are eventually verified and potentially recovered during the process. Nevertheless, to further demonstrate ADG's flexibility and robustness, we conduct additional experiments under more challenging corruption patterns.

Specifically, we explore two extended variants beyond the standard setting:

1. **All-steps corrupted + middle recovery:** all time steps in each trajectory are corrupted, but only the middle element in each slice is recovered, testing whether local recovery remains effective when global context is degraded.

2. **All-steps corrupted + full recovery:** all time steps are corrupted and each is recovered sequentially using ADG, assessing whether full-step recovery can further enhance temporal consistency.

As shown in Table 11, ADG remains effective even when all time steps are corrupted—recovering only the middle element still yields competitive performance, highlighting the strength of local recovery guided by contextual information. Furthermore, performing full-step recovery leads to consistent improvements across all environments, confirming that ADG can scale to dense corruption and flexibly generalize beyond benchmark configurations.

Table 11: Performance under different recovery strategies. Results are averaged over three random seeds.

| Environment | Middle-only Recovery | All-steps Corrupted + Middle Recovery | All-steps Corrupted + Full Recovery |
|---|---|---|---|
| halfcheetah-mr | $31.3 \pm 0.7$ | $26.4 \pm 0.9$ | $33.5 \pm 0.8$ |
| hopper-mr | $45.5 \pm 16.4$ | $40.2 \pm 3.7$ | $52.8 \pm 4.5$ |
| walker2d-mr | $32.7 \pm 4.1$ | $27.9 \pm 3.3$ | $36.1 \pm 2.9$ |

## D.14 Robustness to Non-Additive (Multiplicative) Gaussian Noise

This experiment evaluates the robustness of ADG when the offline dataset is corrupted with multiplicative (non-additive) Gaussian noise. Specifically, each data point $x$ is perturbed as:

$$\tilde{x} = x \cdot (1 + \epsilon), \quad \epsilon \sim \mathcal{N}(0, \sigma^2), \tag{36}$$

where $\epsilon$ represents Gaussian noise scaled by the original value, unlike standard additive noise.

We conducted experiments on standard offline RL benchmark environments under this corruption. Table 12 reports the performance of (i) standard offline RL trained directly on the corrupted dataset, (ii) ADG applied for recovery, and (iii) the original clean dataset. Results are averaged over 4 random seeds.

Table 12: Performance under Multiplicative Gaussian Noise. 'Multiplicative Noise' refers to standard offline RL applied to the corrupted dataset. 'Multiplicative Noise (w/ ADG)' denotes applying ADG for recovery.

| Environment | Multiplicative Noise | Multiplicative Noise (w/ ADG) | Clean |
|---|---|---|---|
| halfcheetah-mr | 6.1±0.4 | 29.8±0.6 | 38.9±0.5 |
| hopper-mr | 12.5±0.9 | 47.2±3.5 | 81.8±6.9 |
| walker2d-mr | 10.7±0.7 | 34.9±2.1 | 59.9±2.7 |

From the results, we observe that while standard offline RL suffers a severe performance drop under multiplicative noise, ADG substantially mitigates this degradation, achieving performance much closer to the clean-data scenario. This demonstrates that ADG is not only robust to additive noise but also effective against non-additive, multiplicative Gaussian corruption, confirming its practical utility in noisy offline RL datasets.

### D.15 Joint Impact of Detection Hyperparameters on Downstream RL Performance

To investigate the joint influence of the detector hyperparameters $k_a$ and $\zeta$ on downstream reinforcement learning performance, we conducted additional experiments on the kitchen-complete dataset under the Random State Attack setting. Specifically, $k_a$ and $\zeta$ were varied over the sets $\{15, 30, 50\}$ and $\{0.10, 0.20, 0.50\}$, respectively. For each configuration, both the detector and denoiser were retrained, and we evaluated the resulting false negative (FN) and false positive (FP) rates, together with the downstream performance of IQL and DT. All results are averaged over three random seeds.

Table 13: Ablation on the trade-offs between FP/FN and downstream RL performance under *Random State Attack*. Results are averaged over three random seeds.

| $k_a$ | $\zeta$ | FN(%) | FP(%) | IQL | DT |
|---|---|---|---|---|---|
|  | 0.10 | 2.7 | 52.6 | 22.0 | 13.1 |
| 15 | 0.20 | 7.3 | 12.0 | 49.9 | 56.4 |
|  | 0.50 | 27.2 | 2.7 | 34.1 | 42.6 |
|  | 0.10 | 1.4 | 45.5 | 25.0 | 12.7 |
| 30 | 0.20 | 3.8 | 4.9 | 45.0 | 12.4 |
|  | 0.50 | 25.4 | 1.7 | 36.0 | 42.1 |
|  | 0.10 | 8.4 | 55.4 | 16.5 | -0.3 |
| 50 | 0.20 | 19.5 | 6.0 | 37.6 | 46.9 |
|  | 0.50 | 29.8 | 1.8 | 18.2 | 31.0 |

As shown in Table 13, the detector achieves the lowest FN at moderate values of $k_a$ and $\zeta$, while both smaller and larger values lead to increased FN, reflecting under- and over-filtering, respectively. Increasing $\zeta$ treats a larger portion of data as clean, which substantially reduces FP but can also increase FN due to missed corruptions.

For downstream performance, we observe a clear negative correlation between FN and final RL results: higher FN values consistently degrade the performance of both IQL and DT. In particular, when $\zeta = 0.50$, both algorithms exhibit significant drops, likely because excessive data filtering removes crucial sequential dependencies. This effect is more pronounced on the relatively small `kitchen-complete` dataset compared to larger benchmarks such as MuJoCo. Interestingly, lower FP does not necessarily yield better downstream results, corroborating our main findings.

## E  Discussion and Limitation

While ADG demonstrates promising results in handling corrupted offline RL datasets, our work has several limitations that warrant discussion. The three-stage diffusion process (detection + denoising) introduces additional training time compared to standard offline RL methods. The performance of

ADG depends on the choice of key hyperparameters such as the temporal slice size $H$ and corruption threshold $\zeta$. The trajectory-based recovery mechanism (Section 4.4) relies on temporal consistency. For environments with highly discontinuous dynamics or sparse rewards, the current slice-based approach may miss long-range dependencies.

