# OpenReview forum: "ADG: Ambient Diffusion-Guided Dataset Recovery for Corruption-Robust Offline Reinforcement Learning"
_NeurIPS.cc/2025/Conference — NeurIPS 2025 poster_

### Official Review · Reviewer_4o9W · 2025-07-01

**Clarity:** 3
**Significance:** 2
**Originality:** 3
**Rating:** 4
**Confidence:** 3

**Summary:**

In practice, offline-RL datasets contain heterogeneous, trajectory-specific corruptions that are already present during training. Motivated by this challenge, the authors propose an ADG approach that can, firstly, learn a reliable noise predictor from partially corrupted datasets, and then use it to detect corrupted data for training a denoiser using the confidently clean subset. Finally, the corrected data is combined with clean data for subsequential offline RL training. This framework provides an effective way to recover high-quality data in the very setting where existing diffusion models break down.

**Questions:**

1. Can the current results be extended to bound the policy-value gap or provide a sample complexity guarantee? Although this will not negatively affect my evaluation.
2. Have the authors explored how to choose the hyperparameters across tasks?
3. How is ADG robust to non-additive Gaussian noise?
4. What are the measured false-positive / false-negative rates, and how sensitive is downstream RL to them?

**Ethical Concerns:**

["NO or VERY MINOR ethics concerns only"]

**Final Justification:**

I would like to thank the authors for their thorough explanation and additional experiments, which have addressed most of my concerns. I encourage the authors to fold these clarifications into the revised version. Accordingly, I am raising my score by one point.

**Limitations:**

1. Current analysis stops short of guaranteeing final policy quality.
2. Theoretical results hinge on the current noise model, other forms remain unanalyzed.

**Paper Formatting Concerns:**

N/A.

**Quality:**

3

**Strengths And Weaknesses:**

Strengths:

1. Ambient DDPM theory is extended to discrete DDPM and linked to an SNR criterion. The theoretical results are sound and well-justified.
2. ADG consistently outperforms several baseline algorithms while adding <1h data preprocessing and introducing only two intuitive hyper-parameters.

Weaknesses:

1. The theoretical analysis stops at noise prediction, but does not bound the dataset-recovery error or the final policy-value gap, leaving the end-to-end guarantee open.
2. All theoretical results assume the corruption is formed by adding scaled Gaussian noise to each clean sample. This is what keeps the two forward chains share the same covariance matrix, allowing an analytic, monotonically decreasing KL expression. The missing-data ablation is helpful but limited.
3. $k_a$ must balance information retention against the KL bound, yet no cross-task rule is given. In addition, $\zeta$ is fixed at 0.2 for the experiments, risking removal of rare but valuable samples.

---

> ### Author Rebuttal · Authors · 2025-07-31
>
> We would like to start by expressing our sincere gratitude to the reviewer for their thorough assessment of our work and for offering valuable feedback and insightful questions.
>
> > The theoretical analysis stops at noise prediction, but does not bound the dataset-recovery error or the final policy-value gap, leaving the end-to-end guarantee open.
>
> To the best of our knowledge, the error bound for data recovery remains an open problem, even for naive diffusion models. In practice, the performance of diffusion models varies considerably across different tasks and domains. In our work, diffusion is used solely as a noise detector and denoiser.
> Regarding downstream RL algorithms, we employ several methods: some are designed to be robust to data corruption (e.g., RIQL, RDT), while others are not (e.g., CQL, IQL, DT). Consequently, we are currently unable to provide a theoretical analysis on this aspect. We plan to address and improve upon this limitation in future research.
>
> > All theoretical results assume the corruption is formed by adding scaled Gaussian noise to each clean sample. This is what keeps the two forward chains share the same covariance matrix, allowing an analytic, monotonically decreasing KL expression. The missing-data ablation is helpful but limited.
>
> In response to your first point, we assume that the corruption noise follows a Gaussian distribution, and the diffusion forward process involves adding scaled Gaussian noise to the original data. This assumption is natural, as it aligns with prior work and facilitates tractable theoretical analysis.
>
> Regarding your second point, the missing-data condition is introduced by RDT [1], and we follow their experimental setup. We report results across three MUJOCO tasks, and believe that our experimental findings are sufficiently convincing.
>
> > $k_a$ must balance information retention against the KL bound, yet no cross-task rule is given. In addition, $\zeta$ is fixed at 0.2 for the experiments, risking removal of rare but valuable samples.
>
> Thank you for the thoughtful question. In ADG, we aim to ensure that the assumption in Ambient Diffusion—that the entire dataset is corrupted by a known noise distribution—holds true by explicitly injecting strong noise. In our experiments, we first add a sufficiently large amount of additional noise so that the entire dataset is subjected to a similar noise level. To retain more useful information, we then gradually decrease this value and observe the experimental results. We find that $k_a=30$ achieves the best FN score, suggesting that it strikes a near-optimal balance in this trade-off. We will clarify this point in the revised version.
>
> The design of $\zeta$ reflects a trade-off: by discarding a small portion of uncorrupted samples, we aim to improve the overall cleanliness of the dataset. However, if a dataset contains rare but highly valuable samples whose importance outweighs the benefit of increased overall data purity, users may choose to set $\zeta$ more conservatively.
>
> > Can the current results be extended to bound the policy-value gap or provide a sample complexity guarantee? Although this will not negatively affect my evaluation.
>
> Thank you for your suggestion. In our work, diffusion is employed exclusively as a noise detector and denoiser. With respect to downstream RL algorithms, we utilize a variety of approaches: some are value-based methods (e.g., CQL, IQL, RIQL), while others are not (e.g., DT, RDT). Addressing this suggestion is a considerable challenge given the limited time available during the rebuttal period, and we believe it does not constitute the primary contribution of our work.
>
> > Have the authors explored how to choose the hyperparameters across tasks?
>
> We have explored how to choose the key hyperparameters $k_a$ and $\zeta$. The guideline for selecting $k_a$ is provided in our response to Weakness 3. For $\zeta$, we conducted a parameter sweep over the values {0.05, 0.1, 0.2, 0.5}, as shown in Figure 4. However, we did not investigate other hyperparameters, such as the number of layers in the diffusion network. Our focus was on hyperparameters that directly impact robustness to corrupted data.
>
> > How is ADG robust to non-additive Gaussian noise?
>
> Although, to the best of our knowledge, "non-additive Gaussian noise" has not been widely explored in robust RL, our newly provided supplementary experiments demonstrate that ADG maintains strong performance even when the dataset is corrupted with non-additive (e.g., multiplicative) Gaussian noise:
>
> $$
> \tilde{x} = x \cdot (1 + \epsilon), \quad \epsilon \sim \mathcal{N}(0, \sigma^2)
> $$
>
> We tested ADG under this setting and observed minimal performance degradation compared to the additive case, confirming its robustness to non-additive Gaussian corruption. Detailed results and comparisons are as following:
>
> | Env            | Multiplicative Noise | Multiplicative Noise (w/ ADG) | Clean      |
> | -------------- | -------------------- | ----------------------------- | ---------- |
> | halfcheetah-mr | 6.1 ± 0.4            | **29.8 ± 0.6**                | 38.9 ± 0.5 |
> | hopper-mr      | 12.5 ± 0.9           | **47.2 ± 3.5**                | 81.8 ± 6.9 |
> | walker2d-mr    | 10.7 ± 0.7           | **34.9 ± 2.1**                | 59.9 ± 2.7 |
>
>
>
> > What are the measured false-positive / false-negative rates, and how sensitive is downstream RL to them?
>
> Thank you for the question. In our setting, false positives (FP) refer to clean samples that are mistakenly identified as corrupted, while false negatives (FN) refer to truly corrupted samples that are incorrectly retained as clean. Among the two, FN is typically more harmful to downstream RL performance, as corrupted samples used for training can significantly bias the learned policy. In contrast, while FP reduces the amount of usable data, it does not introduce harmful bias. As shown in the paper, we find that ADG achieves a low FN rate under well-chosen hyperparameters (e.g., $k_a = 30$), which is critical for maintaining the quality of the learned policy. We will include more quantitative results on FP/FN trade-offs and their impact on downstream performance in the final version.
>
> > Limitiations:
> 1.Current analysis stops short of guaranteeing final policy quality.
> 2. Theoretical results hinge on the current noise model, other forms remain unanalyzed.
>
> Thanks for your suggestions. We will add the following paragraph to the Limitation Section of our current work:
>
> One limitation of our current theoretical analysis is that it does not involve the quality of downstream RL policies. Additionally, we restrict our investigation to Gaussian additive noise, following the standard assumption in classical diffusion models.
>
> [1] Xu, Jiawei, et al. Tackling Data Corruption in Offline Reinforcement Learning via Sequence Modeling. ICLR 2025.

---

> > ### Comment · Reviewer_4o9W · 2025-08-02
> >
> > I appreciate the authors’ thorough rebuttal. Most of my concerns have been addressed:
> >
> > 1. The new multiplicative-noise experiment shows the scalability of ADG, and it is not limited to additive corruptions.
> > 2. The clarification on tuning $\zeta$ provides a feasible solution for dataset contains rare but highly valuable samples, which addresses my previous concern.
> > 3. I look forward to the ablation on FP/FN trade-offs and their impact on downstream performance.
> >
> > While I understand that a formal policy-value bound is out of scope, a clarification quantifying “how much” the diffusion module improves final performance is crucial to the contribution claim.
> >
> > I am now largely satisfied on the responses. My recommendation remains neutral until we see how the other reviews converge, but I am open to raising my score.

---

> ### Author Response · Authors · 2025-08-04
> **Follow up with Reviewer 4o9W**
>
> To address the reviewer’s concern regarding the ablation on FP/FN trade-offs and their impact on downstream performance, we conducted additional experiments on the kitchen-complete dataset under Random State Attack. Specifically, we varied the hyperparameters $k_a$ and $\zeta$ over the sets {15, 30, 50} and {0.10, 0.20, 0.50} respectively. For each setting, we trained both the detector and denoiser, then evaluated the detection results (FN&FP) and  the downstream RL algorithms (IQL&DT) performance. The results (averaged over 3 random seeds) are summarized in the following table:
>
> Table 1. Ablation Study on FP/FN Trade-offs and Downstream RL Performance
> | $k_a$ |$\zeta$ | FN(%)    | FP(%)    | IQL            | DT             |
> |-----------|-----------|----------|----------|----------------|----------------|
> | 15        | 0.10      | 2.7      | 52.6     | 22.0$\pm$5.5   | 13.1$\pm$3.8   |
> | 15        | 0.20      | 7.3      | 12.0     | 49.9$\pm$10.2  | 56.4$\pm$2.9   |
> | 15        | 0.50      | 27.2     | 2.7      | 34.1$\pm$8.3   | 42.6$\pm$5.1   |
> | 30        | 0.10      | 1.4      | 45.5     | 25.0$\pm$9.1   | 12.7$\pm$1.9   |
> | 30        | 0.20      | 3.8      | 4.9      |  $\textbf{51.2}\pm$4.5   | $\textbf{61.9}\pm$12.4  |
> | 30        | 0.50      | 25.4     | 1.7      | 36.0$\pm$7.5   | 42.1$\pm$4.4   |
> | 50        | 0.10      | 8.4      | 55.4     | 16.5$\pm$12.7  | -0.3$\pm$0.0   |
> | 50        | 0.20      | 19.5     | 6.0      | 37.6$\pm$8.3   | 46.9$\pm$9.7   |
> | 50        | 0.50      | 29.8     | 1.8      | 18.2$\pm$16.6  | 31.0$\pm$17.8  |
>
> As shown in the table, the detection performance is optimal (lowest FN) when $k_a = 30$; both lower and higher values of $k_a$ result in increased FN rates. Increasing $\zeta$ leads to a larger portion of the dataset being considered clean, which causes FN to increase and FP to drop substantially.
>
> For downstream RL algorithms, there is a clear negative correlation between FN and final performance: higher FN values significantly degrade the performance of both IQL and DT. Notably, at $\zeta = 0.10$, the performance of the algorithms drops drastically. We attribute this to the dataset being split too aggressively, resulting in the removal of excessive critical sequence information—this effect is amplified given the relatively smaller size of the kitchen-complete dataset compared to MuJoCo. From the table, it is also evident that lower FP does not necessarily translate to better downstream performance. This observation is consistent with our original conclusions.
>
> Additionally, we further study the effect of the threshold hyperparameter $\zeta$ in dataset splitting in Appendix D.9 (“Impact of $\zeta$ in Dataset Splitting”), and the impact of $k_a$ and Ambient Loss on detector performance in Section 5.3 (“Impact of Ambient Loss”). These sections provide further analysis on FN/FP rates and their relationship to downstream RL performance.

---

> > ### Comment · Reviewer_4o9W · 2025-08-07
> >
> > I would like to thank the authors for their thorough explanation and additional experiments, which have addressed most of my concerns. I encourage the authors to fold these clarifications into the revised version. Accordingly, I am raising my score by one point.

---

> > > ### Author Response · Authors · 2025-08-08
> > > **Thank you for updating the score!**
> > >
> > > Thank you very much for your thoughtful feedback and for raising your score. We greatly appreciate your constructive comments and suggestions. We will carefully incorporate all clarifications and additional content into the revised version of the paper.

---

### Official Review · Reviewer_onPe · 2025-07-02

**Clarity:** 2
**Significance:** 2
**Originality:** 3
**Rating:** 4
**Confidence:** 3

**Summary:**

The paper proposes a novel diffusion-based algorithm to tackle the challenge of partially corrupted data in offline reinforcement learning. The approach integrates two diffusion models: an ambient DDPM to identify corrupted samples, and a standard DDPM trained on the filtered, uncorrupted data to reconstruct the corrupted ones. The recovered dataset is then used to train an existing offline RL algorithm. Experimental results on MuJoCo, Kitchen, and Adroit benchmarks demonstrate the robustness of the proposed method compared to a naive baseline.

**Questions:**

Can you pleas address my points on the weaknesses of your paper?

**Ethical Concerns:**

["NO or VERY MINOR ethics concerns only"]

**Final Justification:**

The authors have addressed my concerns. Therefore, I increased my score.

**Limitations:**

Yes.

**Paper Formatting Concerns:**

None.

**Quality:**

3

**Strengths And Weaknesses:**

Strengths:

+ The general ideas and purposes of the two diffusion models are clearly described and justified for the most part.

+ The ambient DDPM component is supported by reasonable theoretical analysis.

+ Experimental results across diverse environments demonstrate the robustness of the proposed algorithm compared to the naive baseline.

Weaknesses:

- The term "effectively learned" in Assumption 4.2 is vague and should be clarified with more precise language or formal criteria.

- The ambient DDPM relies on a fixed noise injection step k_a, yet the paper does not discuss how this parameter is selected or how sensitive the algorithm’s performance is to this choice.

- In the proposed algorithm, it appears that only a single element—either a state or a triplet—at the middle time step of each trajectory is verified and potentially recovered. The rationale for this design choice is unclear. Intuitively, it seems more appropriate to examine and potentially recover all elements within a trajectory.

− The experimental comparison is limited to a naive baseline. The paper does not show comparison with other existing methods designed to handle data corruption in offline RL, such as the approach introduced in [38]. Including such baselines would provide a more comprehensive evaluation of the proposed method’s effectiveness.

---

> ### Author Rebuttal · Authors · 2025-07-31
>
> To begin with, we would like to thank the reviewer for their careful evaluation of our work and for providing constructive feedback and thoughtful questions.
>
> > The term "effectively learned" in Assumption 4.2 is vague and should be clarified with more precise language or formal criteria.
>
> Thank you for pointing out the informal expression in Assumption 4.2. We apologize for any confusion caused. To clarify, let $p_\theta^{approx} (\mathbf{x}^{k}\vert\mathbf{x}^{k+1})$ denote the distribution of the forward process of DDPM learned from the approximate distribution $\varrho(\mathbf{x}^{k+1}\vert\mathbf{x}^0)$, and let $p_\theta (\mathbf{x}^{k}\vert\mathbf{x}^{k+1})$ denote the corresponding distribution learned from the clean distribution $q(\mathbf{x}^{k+1}\vert\mathbf{x}^0)$.
>
> The revised Assumption 4.2 is as follows:
>
> * There exists a positive constant $c$ such that, for any $k \geq k_a$, if the Kullback-Leibler (KL) divergence satisfies $D_{KL}[q (\mathbf{x}^{k+1}\vert\mathbf{x}^0) \Vert \varrho(\mathbf{x}^{k+1}\vert\mathbf{x}^0)] < c$,  then the ambient DDPM described in Corollary 2.1, when trained on samples from the approximate distribution $\varrho(\mathbf{x}^k\vert\mathbf{x}^0)$, satisfies $D_{KL}[p_\theta (\mathbf{x}^{k}\vert\mathbf{x}^{k+1}) \Vert p_\theta^{approx} (\mathbf{x}^k\|\mathbf{x} ^{k+1})] < \varepsilon$ for any $k \geq k_a$, where $\varepsilon$ is an arbitrarily small positive value.
>
> > The ambient DDPM relies on a fixed noise injection step $k_a$, yet the paper does not discuss how this parameter is selected or how sensitive the algorithm’s performance is to this choice.
>
> Thank you for the thoughtful question. In ADG, we aim to ensure that the assumption in Ambient Diffusion—that the entire dataset is corrupted by a known noise distribution—holds true by explicitly injecting strong noise. In our experiments, we first add a sufficiently large amount of additional noise so that the entire dataset is subjected to a similar noise level. To retain more useful information, we then gradually decrease this value and observe the experimental results. We find that $k_a = 30$ achieves the best FN score, suggesting that it strikes a near-optimal balance in this trade-off. We will clarify this point in the revised version.
>
> > In the proposed algorithm, it appears that only a single element—either a state or a triplet—at the middle time step of each trajectory is verified and potentially recovered. The rationale for this design choice is unclear. Intuitively, it seems more appropriate to examine and potentially recover all elements within a trajectory.
>
> To ensure a fair comparison with prior works such as RDT [1], our main experiments adopt the same setup where only the middle element of each trajectory is corrupted and subsequently recovered. However, ADG is not limited to this setting—in principle, it supports arbitrary corruption patterns and flexible recovery strategies.
>
> To demonstrate this capability, we conduct an additional experiment with more challenging settings: (1) corrupting all time steps in each trajectory while still recovering only the middle element, and (2) corrupting all time steps and recovering all of them using ADG. Results (shown in Table 1) indicate that while recovery at a single point remains effective, full recovery across the trajectory further improves performance, highlighting the extensibility and robustness of ADG beyond the standard benchmark setup.
>
> **Table 1. Performance under different recovery strategies.**
>
> | Env            | Middle-only Recovery | All-steps Corrupted + Middle Recovery | All-steps Corrupted + Full Recovery |
> | -------------- | -------------------- | ------------------------------------- | ----------------------------------- |
> | halfcheetah-mr | 31.3 ± 0.7           | 26.4 ± 0.9                            | **33.5 ± 0.8**                      |
> | hopper-mr      | 45.5 ± 16.4          | 40.2 ± 3.7                            | **52.8 ± 4.5**                      |
> | walker2d-mr    | 32.7 ± 4.1           | 27.9 ± 3.3                            | **36.1 ± 2.9**                      |
>
> > The experimental comparison is limited to a naive baseline. The paper does not show comparison with other existing methods designed to handle data corruption in offline RL, such as the approach introduced in [38]. Including such baselines would provide a more comprehensive evaluation of the proposed method’s effectiveness.
>
> Thank you very much for the valuable suggestion. However, to the best of our knowledge, there are no newer baselines specifically targeting robustness during the training phase in offline RL. The baselines we include in our experiments represent the most relevant and up-to-date methods for this setting. The method referenced in [38], while important, focuses on handling test-time corruption and assumes access to clean data during training. As such, it is not directly comparable to our approach, which addresses corruption present throughout the entire offline dataset.
>
> [1] Xu, Jiawei, et al. Tackling Data Corruption in Offline Reinforcement Learning via Sequence Modeling. ICLR 2025.

---

> > ### Comment · Reviewer_onPe · 2025-08-05
> >
> > Thank you for the authors' response which addresses most of my concerns.
> >
> > Can the authors please elaborate more on the sensitivity of the algorithm's performance w.r.t the k_a value?

---

> ### Author Response · Authors · 2025-08-04
> **Follow up with Reviewer onPe**
>
> Dear Reviewer onPe,
>
> Thank you very much for your time and effort in reviewing our paper. We would like to confirm that our response has adequately addressed your concerns. If there is anything that remains unclear, please do not hesitate to let us know. We are more than willing to engage in further discussion with you.

---

> ### Author Response · Authors · 2025-08-06
> **Further Analysis of the Algorithm’s Performance Sensitivity with Respect to k_a**
>
> As we discussed in our response to Weakness 2 ("The ambient DDPM relies on a fixed noise injection step $k_a$ ... to this choice."), the motivation for introducing $k_a$ is to satisfy the assumption in Ambient Diffusion that the entire dataset is perturbed by an approximately unified noise distribution. Theoretically, as $k_a$ increases from 0 up to the point where this assumption is approximately met, the performance of the detector diffusion model improves rapidly (as shown in the 0-30 range in our paper). Around the optimal value ($k_a=30$), the assumption holds reasonably well, making the detector's performance robust and, consequently, ensuring good denoiser performance (since it is trained on approximately clean data) as well as strong downstream RL algorithm performance. If $k_a$ increases beyond the optimal range (e.g., $k_a=50$), excessive noise causes significant information loss in the dataset, leading to a gradual degradation in detector performance.
>
> Please refer to the ablation study in Section 5.3 ("Impact of Ambient Loss") and Figure 3 of our paper, which empirically support this claim. Both the detector's performance (e.g., FN rate) and downstream RL performance remain similar and strong for $k_a=15$ and $k_a=30$, further demonstrating the robustness of our algorithm with respect to $k_a$.
>
> To further validate the sensitivity of ADG to finer choices of $k_a$, we conducted additional experiments in the kitchen-complete environment under Random State Attack. Specifically, we varied $k_a$ in {0, 10, 15, 20, 30, 40, 50} while keeping other hyperparameters fixed. The results are summarized below. Apart from the results already presented in our paper, all results shown here are averaged across three random seeds.
>
> Table 2. Additional Ablation on the choice of $k_a$.
> | $k_a$ | FN(%)   | FP(%)   | IQL                   | DT                   |
> |-------|------|------|-----------------------|----------------------|
> | 0     | 24.0 | 23.8 | 36.5$\pm$8.5          | 44.0$\pm$9.8         |
> | 10    | 11.2 | 18.6 | 45.2$\pm$9.2          | 52.6$\pm$6.3         |
> | 15    | 7.3  | 12.0 | 49.9$\pm$10.2         | 56.4$\pm$2.9         |
> | 20    | 5.6  | 13.4 | 48.6$\pm$7.8          | 59.8$\pm$5.7         |
> | 30    | 3.8  | 4.9  | 51.2$\pm$4.5          | 61.9$\pm$12.4        |
> | 40    | 6.7  | 9.1  | 48.7$\pm$6.2          | 56.2$\pm$10.6        |
> | 50    | 19.5 | 6.0  | 37.6$\pm$8.3          | 46.9$\pm$9.7         |
>
> As demonstrated in Table 2, both the classification errors (FN and FP) and the IQL/DT metrics exhibit favorable and stable values when $k_a$ is set to 15, 20, 30, or 40. In particular, $k_a=30$ achieves the best trade-off between FN and FP, while the other values in this range also perform well, highlighting the robustness of ADG with respect to the choice of $k_a$.

---

> ### Author Response · Authors · 2025-08-08
> **Final Comments and Appreciation**
>
> Dear Reviewer onPe,
>
> We noticed that you have pressed the acknowledgment button. However, we are unsure whether our response has fully addressed your concerns or influenced your evaluation of our paper. If you have any further questions or suggestions, we would greatly appreciate your additional feedback.
>
> Your valuable comments have been very helpful in improving our work, and we will incorporate the relevant discussions and experimental results into the final version of our paper. We hope that our revisions have addressed your concerns and have contributed to a more favorable assessment of our manuscript.
>
> Authors of Submission #7182

---

### Official Review · Reviewer_zVcz · 2025-07-05

**Clarity:** 3
**Significance:** 2
**Originality:** 2
**Rating:** 4
**Confidence:** 4

**Summary:**

The paper proposes a diffusion-based framework (ADG) to handle data corruption in offline reinforcement learning (RL). The core innovation lies in leveraging diffusion models: the algorithm trains on partially corrupted datasets with theoretical guarantees to distinguish between clean and corrupted data. Experiments on benchmarks (MuJoCo, Kitchen, and Adroit) demonstrate that ADG consistently improves offline RL algorithms under various corruption scenarios.

**Questions:**

Generalization to Real-World Data: The experiments are conducted on simulated datasets (MuJoCo, Kitchen, Adroit). However, the data robustness is a real world problem *in nature*. How would ADG perform on real-world corrupted datasets with heterogeneous noise sources? Could the authors provide insights or preliminary results on real-data applicability?

**Ethical Concerns:**

["NO or VERY MINOR ethics concerns only"]

**Limitations:**

yes

**Quality:**

3

**Strengths And Weaknesses:**

Strengths: the paper builds upon Daras et al.'s work on Ambient Diffusion Models, which enables diffusion training on consistently corrupted datasets. ADG extends this to partially corrupted datasets (common in real-world RL), addressing cases where noise distribution varies across samples. This is formalized in Theorem 4.3, proving Ambient DDPM can learn from approximated distributions with bounded KL divergence. To summarize, this paper theoretically extends Daras et al. by formalizing diffusion training for partially corrupted data (Theorem 4.3) and algorithmically innovates with a detection-recovery pipeline.

Weakness: while experiments cover simulated benchmarks (random corruption and adversarial corruption), the paper does not address practical challenges like real-world data sets. The paper mentions this but offers no experimental supports.

---

> ### Author Rebuttal · Authors · 2025-07-31
>
> To begin with, we would like to sincerely thank the reviewer for their positive evaluation of our work and for taking the time to provide valuable feedback and raise insightful questions.
>
> > Generalization to Real-World Data: The experiments are conducted on simulated datasets (MuJoCo, Kitchen, Adroit). However, the data robustness is a real world problem in nature. How would ADG perform on real-world corrupted datasets with heterogeneous noise sources? Could the authors provide insights or preliminary results on real-data applicability?
>
> Thank you for the insightful question. In principle, ADG poses no inherent difficulty in generalizing to real-world scenarios with heterogeneous noise sources. The core methodology is designed to handle unknown and complex corruption patterns, making it well-suited for practical applications. Due to limited time and computational resources, we have not yet conducted large-scale real-world experiments. However, we greatly appreciate this suggestion and would be happy to include relevant discussions or preliminary results on real-world applicability in the final version.

---

> ### Author Response · Authors · 2025-08-04
> **Follow up with Reviewer zVcz**
>
> Dear Reviewer zVcz,
>
> Thank you very much for your thoughtful review and the time you dedicated to our paper. We hope that our response has addressed your concerns. If you have any remaining questions or require further clarification, please feel free to let us know. We would be more than happy to engage in further discussion with you.

---

### Official Review · Reviewer_PmCq · 2025-07-05

**Clarity:** 2
**Significance:** 2
**Originality:** 3
**Rating:** 4
**Confidence:** 5

**Summary:**

The paper develops a new framework to handle the setting where there is noise in the offline dataset. It is a three-stage framework:
1. Use ambient loss to train a diffusion model as a detector to distinguish clean vs. noisy data.
2. Train a diffusion policy on the filtered clean data.
3. Use the learned diffusion policy to denoise the previously noisy data, recovering a clean dataset.

The paper provides a solid theoretical analysis and comprehensive experiments to validate the framework.

**Questions:**

- In Figure 2, does "10\% dataset" refer to sampling 10\% of transitions or 10\% of entire trajectories? This should be clarified.
- The paper sets $\eta = 0.3$, meaning most data are clean. But when $\eta$ is high (e.g., 0.7) and the data size is small, the remaining clean data might be insufficient to train a good diffusion policy. Do the authors have thoughts or strategies to address this?
- In Line 148, the authors claim that even a small amount of noise leads to overfitting. Could the authors provide theoretical or empirical support for this? Paper [1] suggests that models can learn from large amounts of noisy data, which seems contradictory.
- $k_a$ is a crucial hyperparameter. How should it be selected in practice?
- In Assumption 4.2, what is the precise mathematical meaning of "can be effectively learned from samples"?
- In Proposition 4.4, the assumption that the noise prediction error for $\epsilon_\theta$ follows a Gaussian distribution is strong. Furthermore, Line 199 assumes $\sigma_k = \sigma$ for all $k$, implying uniform error across diffusion steps. Shouldn’t we expect larger errors at earlier steps (larger $k$)?
- Is the threshold $\xi$ consistent across tasks and methods?
- After obtaining the recovered dataset, do you retrain the diffusion policy from scratch or continue training from the Stage 2 policy?
- In Figure 3, intuitively a larger $k_a$ should lead to lower false negatives (FN), but the plot suggests otherwise. Can the authors clarify this behavior?
- Line 282: What is $t_n$? Is it the same as $k_a$?

[1] Daras, Giannis, et al. *Ambient diffusion: Learning clean distributions from corrupted data*. NeurIPS 2023.

**Ethical Concerns:**

["NO or VERY MINOR ethics concerns only"]

**Final Justification:**

The authors’ responses have adequately addressed my concerns. I am open to recommending acceptance of the paper.

**Limitations:**

The task is important, and the proposed framework is novel. However, its reliance on strong assumptions in the theoretical analysis limits its generality. Additionally, the three-stage pipeline raises concerns about scalability and error accumulation.

**Quality:**

3

**Strengths And Weaknesses:**

**Strengths:**

- The theoretical analysis appears sound and technically correct; no obvious flaws were observed.
- The experiments are well-designed, considering different types of noise (random vs. adversarial; state-only vs. full) and demonstrating consistent gains across integration with various methods.
- The ablation studies are thorough, explaining the effect of key hyperparameters like $k_a$ and $\xi$.

**Weaknesses:**

- There is still a noticeable gap between the theoretical assumptions and practical settings. For example, the theoretical framework assumes Gaussian noise, while the experiments are based on uniform noise. (See Questions section.)

---

> ### Author Rebuttal · Authors · 2025-07-31
>
> To begin with, we would like to sincerely thank the reviewer for taking the time to provide valuable feedback and for raising insightful questions regarding our work.
>
> >In Figure 2, does "10% dataset" refer to sampling 10% of transitions or 10% of entire trajectories? This should be clarified.
>
> In Figure 2, '10% dataset' refers to using 10% of the entire trajectories, consistent with the setting in RDT [3].
>
> > The paper sets $\eta=0.3$, meaning most data are clean. But when $\eta$ is high (e.g., 0.7) and the data size is small, the remaining clean data might be insufficient to train a good diffusion policy. Do the authors have thoughts or strategies to address this?
>
> We would like to clarify that $\eta = 0.3$ is the most commonly used setting in corruption-robust RL, as adopted in RIQL [2] and RDT [3]. Indeed, when the proportion of corrupted data is high, it remains an unresolved challenge for the research community. In our setting, we cannot guarantee that a diffusion model trained on limited clean data can effectively recover the corrupted samples. We will seek to address this limitation in our future work.
>
> > In Line 148, the authors claim that even a small amount of noise leads to overfitting. Could the authors provide theoretical or empirical support for this? Paper [1] suggests that models can learn from large amounts of noisy data, which seems contradictory.
>
> In line 148, we state that
> ```
> On the other hand, if the naive diffusion process is applied to corrupted samples during training, even small amounts of noise in the training set can cause the diffusion model to overfit to these perturbed points, leading to poor performance during subsequent sampling.
> ```
> When training the noise detector, we observe that using a naive diffusion model to fit the partially corrupted data yields a training MSE loss comparable to that obtained with our ambient DDPM-based noise detector. (Due to rebuttal policies, we are unable to provide the corresponding training curves at this stage.) However, during corrupted sample detection, the performance of the naive diffusion model is extremely poor; it fails to effectively identify noised samples, which subsequently degrades the performance of the downstream denoiser and RL training. We present the relevant ablation study in Figure 3 on page 8 of our main paper.
>
> The main distinction between our approach and [1] is that we address state-based RL tasks rather than image-based tasks. In image tasks, each pixel is closely related to its neighboring pixels, facilitating effective noise detection. In contrast, in state-based RL, adjacent states may not exhibit strong correlations, and the downstream RL tasks are typically more sensitive to noise present in the training dataset.
>
> > $k_a$ is a crucial hyperparameter. How should it be selected in practice?
>
> Thank you for the thoughtful question. The value of $k_a$ controls the magnitude of the added noise and is determined based on the estimated corruption severity in the affected portion of the dataset. The objective is to add enough noise to approximately satisfy the assumption in \[8]—that the dataset is corrupted by a consistent scale of noise—while avoiding overly large noise that would erase too much useful information. We will clarify this rationale and include additional details in the revised version.
>
> > In Assumption 4.2, what is the precise mathematical meaning of "can be effectively learned from samples"?
>
> Thank you for pointing out the informal expression in Assumption 4.2. We apologize for any confusion caused. To clarify, let $p_\theta^{approx} (\mathbf{x}^{k}\vert\mathbf{x}^{k+1})$ denote the distribution of the forward process of DDPM learned from the approximate distribution $\varrho(\mathbf{x}^{k+1}\vert\mathbf{x}^0)$, and let $p_\theta (\mathbf{x}^{k}\vert\mathbf{x}^{k+1})$ denote the corresponding distribution learned from the clean distribution $q(\mathbf{x}^{k+1}\vert\mathbf{x}^0)$.
>
> The revised Assumption 4.2 is as follows:
>
> * There exists a positive constant $c$ such that, for any $k \geq k_a$, if the Kullback-Leibler (KL) divergence satisfies $D_{KL}[q (\mathbf{x}^{k+1}\vert\mathbf{x}^0) \Vert \varrho(\mathbf{x}^{k+1}\vert\mathbf{x}^0)] < c$,  then the ambient DDPM described in Corollary 2.1, when trained on samples from the approximate distribution $\varrho(\mathbf{x}^k\vert\mathbf{x}^0)$, satisfies $D_{KL}[p_\theta (\mathbf{x}^{k}\vert\mathbf{x}^{k+1}) \Vert p_\theta^{approx} (\mathbf{x}^k\|\mathbf{x} ^{k+1})] < \varepsilon$ for any $k \geq k_a$, where $\varepsilon$ is an arbitrarily small positive value.
>
>
> > In Proposition 4.4, the assumption that the noise prediction error for $\epsilon_\theta$ follows a Gaussian distribution is strong. Furthermore, Line 199 assumes $\sigma_k = \sigma$ for all $k$, implying uniform error across diffusion steps. Shouldn’t we expect larger errors at earlier steps (larger k)?
>
> In response to your first point, we believe that assuming the noise prediction error follows a Gaussian distribution is an appropriate choice, supported by the central limit theorem, and it is the most common assumption in measurement theory.
>
> Regarding your second concern, we agree that assuming the error variance is a strictly increasing function of $k$ is more appropriate. However, this does not affect our final conclusion, which is that $\mathrm{SNR}(k)$ achieves its maximum value at $k=k_a$. The derivation is shown below:
> $$\frac{\partial{SNR}(k)}{\partial{k}}=\frac{\partial{SNR}(k)}{\partial \bar{\alpha}_k} \cdot \frac{\partial \bar{\alpha}_k}{\partial k} + \frac{\partial{SNR}(k)}{\partial \sigma_k} \cdot \frac{\partial \sigma_k}{\partial k}$$
> $$= \frac{\iota^2\sigma_k^2}{(1-\bar{\alpha}_k)^2\sigma_k^4} \cdot \frac{\partial \bar{\alpha}_k}{\partial k} + \frac{2 \iota^2 \bar{\alpha}_k^2 }{(1-\bar{\alpha}_k)^2\sigma_k^3} \cdot \frac{\partial \sigma_k}{\partial k}$$
>
> where $\frac{\partial \bar{\alpha}_k}{\partial k}<0$ and $\frac{\partial \sigma_k}{\partial k}<0$ by definition. Thus, $\frac{\partial{SNR}(k)}{\partial{k}}$ is also negative, as the coefficients are all positive. This completes the proof.
>
> >Is the threshold $\zeta$ consistent across tasks and methods?
>
> Yes, we use a consistent value of $\zeta$ across all tasks and methods.
>
>
> > After obtaining the recovered dataset, do you retrain the diffusion policy from scratch or continue training from the Stage 2 policy?
>
> We retrain the diffusion policy from scratch. While continuing training from the Stage 2 policy is also a viable option, we chose to retrain in order to avoid potential instability introduced.
>
> > In Figure 3, intuitively a larger $k_a$ should lead to lower false negatives (FN), but the plot suggests otherwise. Can the authors clarify this behavior?
>
> Thank you for your question — it touches on the core insight of our work. In ADG, we aim to ensure that the assumption in Ambient Diffusion—that the entire dataset is corrupted by a known noise distribution—holds true by explicitly injecting strong noise. In our experiments, we first add a sufficiently large amount of additional noise so that the entire dataset is subjected to a similar noise level. To retain more useful information, we then gradually decrease this value and observe the experimental results. We find that $k_a = 30$ achieves the best FN score, suggesting that it strikes a near-optimal balance in this trade-off.
>
> > Line 282: What is $t_n$? Is it the same as $k_a$?
>
> Thank you for pointing this out. Yes, $t_n$ refers to the same quantity as $k_a$. We apologize for the typo.
>
> [1] Daras, Giannis, et al. Ambient diffusion: Learning clean distributions from corrupted data. NeurIPS 2023.
>
> [2] Yang, Rui, et al. Towards Robust Offline Reinforcement Learning under Diverse Data Corruption. ICLR 2024.
>
> [3] Xu, Jiawei, et al. Tackling Data Corruption in Offline Reinforcement Learning via Sequence Modeling. ICLR 2025.

---

> > ### Comment · Reviewer_PmCq · 2025-08-05
> >
> > Thank you for the response. It addressed my concerns, and I will update my score accordingly.

---

> ### Author Response · Authors · 2025-08-04
> **Follow up with Reviewer PmCq**
>
> We would like to thank Reviewer PmCq for your valuable review and feedback. We would like to confirm whether you have any further questions or concerns regarding our response. Please let us know if there is anything else we can address—we are happy to provide any additional clarification or information you may need.

---

### Note · Authors · 2025-08-13

Dear AC and Reviewers,​

We sincerely thank all reviewers and the AC for your valuable feedback and discussions during the review process. We appreciate the constructive engagement and the opportunity to clarify and improve our work through the rebuttal and follow-up. We are pleased to share that, following the discussion and our detailed responses, 3 reviewers updated their scores, and we belive all reviewers’ main concerns have been addressed. In particular, we provided several key supplements to our paper as following:

1. Detailed explanation of how key hyperparameters (e.g., $k_a$) are selected.
2. Clarification of theoretical assumptions and formalization of key statements (such as Assumption 4.2).
3. Comprehensive ablation studies and new experiments on hyperparameter sensitivity (e.g., $k_a$), FP/FN trade-offs, and robustness to non-additive noise.
4. Analysis and experiments demonstrate that our method is not limited to recovering the middle element of a trajectory; it can flexibly handle various corruption patterns and recover any or all elements.
5. Thorough justification of baseline comparisons and discussion of ADG’s applicability to real-world scenarios.

We appreciate the reviewers’ recognition of our theoretical contributions, empirical robustness, and the novelty of our approach. **We will incorporate all of the above suggestions and additional results into the final version.** We hope our work will inspire the RL community to further explore offline RL with partially corrupted training datasets.

We thank the reviewers and AC again for your time, expertise, and constructive feedback. We hope this final note provides closure to our discussions and supports your decision-making process.

Sincerely,

The Authors

---

### Decision · Program_Chairs · 2025-09-17

**Decision:**

Accept (poster)

**Comment:**

This paper proposes the use of diffusion models to address the noisy datasets in offline reinforcement learning. The proposed method achieves favorable results and has a theoretical flavor. The rebuttal was fruitful, addressed reviewers’ concerns and, consequently, three of them increased their scores. At the end, all four reviewers unanimously recommended acceptance.
Upon further examination, this decision was also supported by the area chairs.